# Phosphorylation of MAVS/VISA by Nemo-like kinase (NLK) for degradation regulates the antiviral innate immune response

Shang-Ze Li [ID] [1,2,6], Qi-Peng Shu[1,6], Yang Song[1], Hui-Hui Zhang[3], Yi Liu[1], Bing-Xue Jin[1], Tian-Zi Liuyu[4], Chao Li[1], Xi-Chen Huang[1], Run-Lei Du[1], Wei Song[5], Bo Zhong[4] & Xiao-Dong Zhang [ID] [1,2]

MAVS is essential for antiviral immunity, but the molecular mechanisms responsible for its tight regulation remain poorly understood. Here, we show that NLK inhibits the antiviral immune response during viral infection by targeting MAVS for degradation. NLK depletion promotes virus-induced antiviral cytokine production and decreases viral replication, which is potently rescued by the reintroduction of NLK. Moreover, the depletion of NLK promotes antiviral effects and increases the survival times of mice after infection with VSV. NLK interacts with and phosphorylates MAVS at multiple sites on mitochondria or peroxisomes, thereby inducing the degradation of MAVS and subsequent inactivation of IRF3. Most importantly, a peptide derived from MAVS promotes viral-induced IFN-β production and antagonizes viral replication in vitro and in vivo. These findings provide direct insights into the molecular mechanisms by which phosphorylation of MAVS regulates its degradation and influences its activation and identify an important peptide target for propagating antiviral responses.

[1] Hubei Key Laboratory of Cell Homeostasis, College of Life Sciences, Wuhan University, Wuhan 430072 Hubei, China. [2] Medical Science Research Center, Zhongnan Hospital of Wuhan University, Wuhan 430071 Hubei, China. [3] College of Medicine, Hunan Normal University, Changsha 410013 Hunan, China. [4] College of Life Sciences, Wuhan University, Wuhan 430072 Hubei, China. [5] Department of Biochemistry and Molecular Biology, State Key Laboratory of Medical Molecular Biology, Institute of Basic Medical Sciences Chinese Academy of Medical Sciences, Peking Union Medical College, Beijing 100005, China. [6] These authors contributed equally: Shang-Ze Li, Qi-Peng Shu. Correspondence and requests for materials should be addressed to S.-Z.L. (email: shangze. li@whu.edu.cn) or to W.S. (email: songwei@ibms.pumc.edu.cn) or to B.Z. (email: zhongbo@whu.edu.cn) or to X.-D.Z. (email: zhangxd@whu.edu.cn)

The innate immune response is the first line of host defense against pathogen infection. Pathogen-associated molecular patterns are detected by several classes of pattern recognition receptors that initiate innate immune responses, including Toll-like receptors (TLRs), RIG-I-like receptors, Nod-like receptors (NLRs), and cytoplasmic DNA sensors[1–3]. After pathogen recognition, pattern recognition receptors trigger the nuclear translocation of the nuclear factor (NF)-κB transcription factor, production of type I interferons (IFNs), and induction of subsequent adaptive immune responses[4,5]. RIG-I, MDA5, and TLR3 play important roles in RNA virus recognition and subsequent TRAF3 activation. In endosomes, TLR3 recognizes viral double-stranded RNA and triggers a signaling pathway mediated by the adapters TRIF and TRAF3[6,7]. In the cytoplasm, the RNA helicases RIG-I and MDA5 function as cytoplasmic RNA sensors that can recognize both the Sendai virus (SeV) and vesicular stomatitis virus (VSV) RNA viruses, resulting in recruitment of the mitochondrial antiviral-signaling protein (MAVS; VISA, IPS-1, or Cardif) and TRAF3-dependent assembly of K63-linked polyubiquitin chains, which further activates the downstream signaling molecules TBK1 and IRF3, to achieve the type I interferon response after ligand recognition[8,9]. In addition, dA:dT-rich dsDNA is reportedly transcribed into dsRNA by DNA-dependent RNA polymerase III (Pol III), which in turn stimulates RIG-I and its downstream signaling cascade[10,11].

Mitochondria provide most of the energy for cells and are therefore referred to as the powerhouse of the cell. MAVS was first identified by four independent groups in 2005[12–15]. The mitochondrial location of MAVS is critical for the antiviral response; thus, depending on the mitochondrial location and virus stimulation, MAVS forms prion-like aggregates and recruits upstream and downstream proteins into the mitochondria as the MAVS regulome to transmit the antiviral signal[16]. Therefore, mitochondria are considered a platform and hub for antiviral innate immune signaling. Posttranslational modification (PTM) is an important regulatory mechanism for signaling, and the ubiquitination of MAVS has been extensively studied. RNF5 has been reported to shut down antiviral responses through the ubiquitination and degradation of MAVS[17]. Another E3 ligase, AIP4, mediates K48-linked polyubiquitination through PCBP2, leading to the degradation of MAVS[18]. Phosphorylation is a critical signaling process for cell biological behavior. Although three kinases, PLK1 and c-Abl and PKACs, were found to be associated with MAVS and to orchestrate virus-induced MAVS signaling[19–21], whether MAVS is the real substrate of these three kinases in response to viral stimulation remains elusive. Furthermore, the association between the phosphorylation and ubiquitination of MAVS urgently requires determination.

As uncontrolled immune responses can cause extensive damage to the host, NF-κB activation and type I interferon signaling must be tightly regulated to maintain host immune homeostasis[22]. Despite the importance of RIG-I-like receptor (RLR)-mediated type I interferon signaling to the immune response, the molecular mechanisms responsible for the regulation of RLR-mediated signaling via the posttranscriptional modification of MAVS remain poorly understood.

Nemo-like kinase (NLK), the vertebrate homolog of the nemo gene, was discovered in mice by Brott et al.[23]. NLK can directly phosphorylate transcription factors or signaling pathway intermediates, exerting either positive or negative effects depending on the NLK target. The serine/threonine protein kinase activity of NLK was initially reported to negatively regulate the Wnt signaling pathway by phosphorylating TCF/LEF factors and inhibiting interactions between the β-catenin-TCF complex and DNA[24]. Subsequently, a number of transcriptional regulators were identified as substrates of NLK. For example, NLK phosphorylates c-Myb, resulting in its degradation[25]. NLK-mediated Foxo1 phosphorylation has been shown to inhibit Foxo1-mediated transcription by promoting its nuclear export[26]. STAT3 phosphorylation induced by the TAK1-NLK cascade is indispensable for TGF-β-mediated mesoderm induction during early Xenopus development[27]. NLK also regulates Wnt-5a signaling by phosphorylating SETDB1[28]. Recent studies have shown that NLK acts as a negative regulator of Notch signaling by phosphorylating Notch1-ICD and interfering with the formation of active Notch transcriptional complexes[29]. In addition, NLK is involved in nervous system development[30,31] and cancer cell proliferation[32–34]. Our group has shown that NLK regulates NF-kappa B signaling by disrupting the interaction of TAK1 with IKK[35]. These findings imply potential roles of NLK in regulating immune responses.

Here, we sought to characterize the function of NLK in the innate immune response, and we report the potent negative regulation of type I interferon signaling by NLK. NLK strongly hinders type I interferon production by phosphorylating MAVS, thereby resulting in its degradation and subsequent inactivation of downstream signaling pathways. A new peptide derived from MAVS was characterized against viruses, revealing obvious antiviral effects both in vitro and in vivo. Our data suggest that NLK is vital for the cellular homeostatic control of innate immunity and identify a new peptide targeting the NLK/MAVS complex for the propagation of antiviral responses.

## Results

**NLK is a negative regulator of virus-induced signaling.** To characterize the kinase that potentially regulates virus-induced signaling, we screened ~100 kinases using an IFN-β luciferase reporter after SeV stimulation. NLK significantly inhibited SeV-induced IFN-β luciferase activity (Supplementary Fig. 1a, b). Although NLK is not the only kinase that may orchestrate SeV-induced IFN-β luciferase activation, further confirmation indicated that NLK showed the strongest inhibitory effect. To further verify the function of NLK in this pathway, we cotransfected human embryonic kidney cells (HEK293T cells) with an IFN-β luciferase reporter plasmid and increasing concentrations of the NLK expression plasmid and then treated the cells with SeV to trigger type I interferon signaling. NLK potently inhibited SeV-induced IFN-β luciferase reporter activation (Fig. 1a) in a dose-dependent manner, whereas IFN-γ-induced activation of the IRF1 promoter had no effect during NLK overexpression (Fig. 1b). Because IFN-β activation requires coordination between the activation of NF-κB and IRF3, we used an interferon-stimulated response element (ISRE) luciferase reporter that required only IRF3 activation to evaluate whether the NLK-dependent inhibition of type I interferon was dependent on its inhibitory effect on ISRE signaling. SeV-induced ISRE luciferase reporter activity was potently inhibited by NLK, suggesting that NLK inhibited IFN-β activation by blocking IRF3 signaling (Fig. 1c). To further determine the roles of NLK in viral-mediated signaling, we next performed real-time PCR experiments. We demonstrated that NLK inhibited the SeV-induced transcription of IFN-β, ISG15, ISG56, and RANTES mRNA (Fig. 1d).

To determine whether the mRNA and protein levels of NLK were altered in response to SeV stimulation, we detected both the mRNA and protein levels of NLK at the indicated time after exposure to SeV, revealing that these levels were not obviously changed (Supplementary Fig. 1c, d). We also detected the NLK protein levels in various cell lines to assess whether they were higher in immune system cells than in other cell lines. We chose to perform immunoblot experiments with TPH-1 human monocytes, RKO colon carcinoma cells, N87 gastric carcinoma

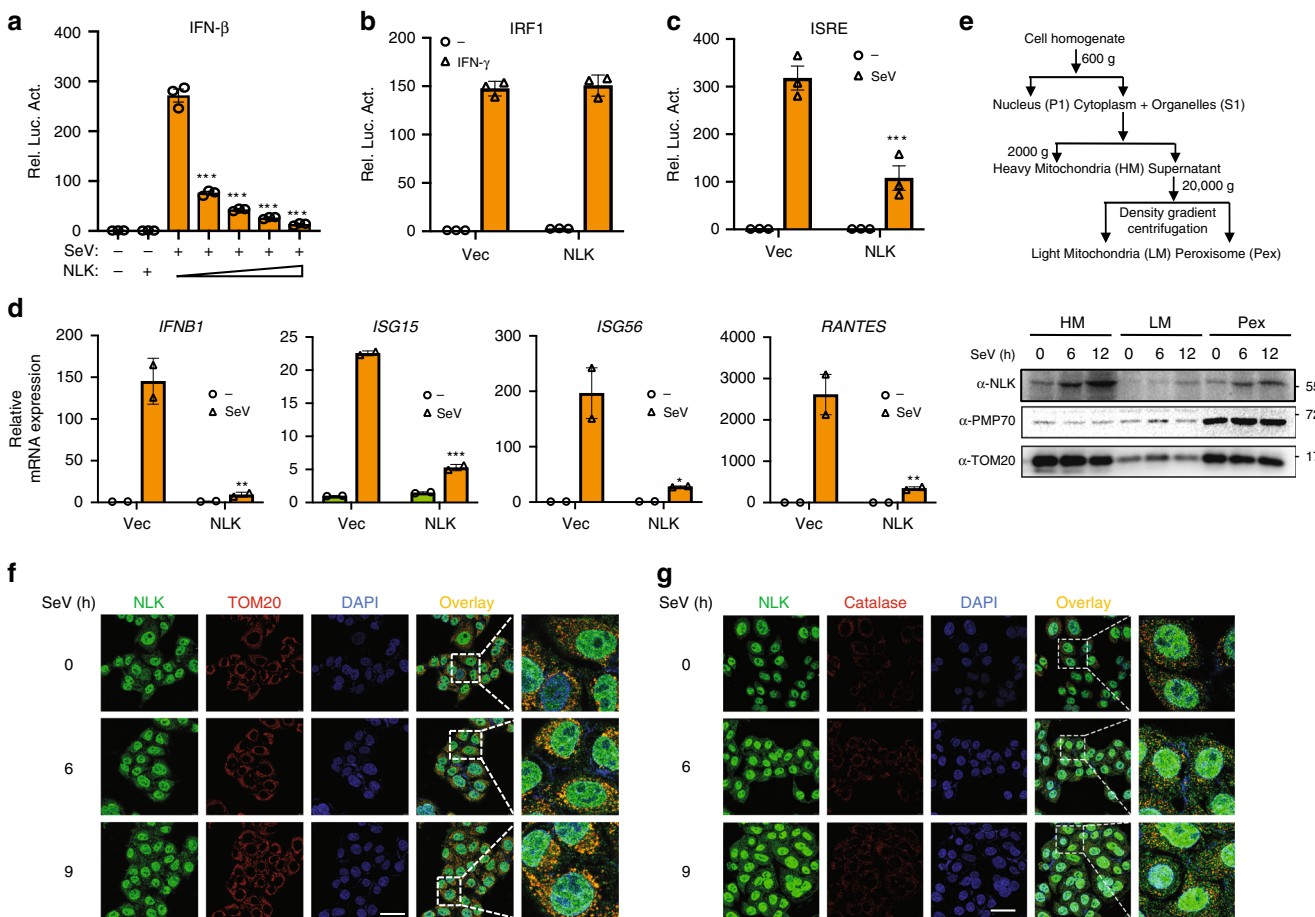

**Fig. 1** NLK negatively regulates the type I interferon signaling pathway. **a** NLK expression inhibits SeV-triggered IFN-β signaling in a dose-dependent manner. HEK293T cells were cotransfected with the IFN-β reporter (100 ng) and increasing concentrations of the Flag-NLK expression plasmid (0, 50, 100, 200, or 400 ng). After 24 h, SeV was added to the cells for 12 h, and reporter gene activity was assayed using a luciferase kit. **b** NLK does not inhibit the IFN-γ-induced activation of the IRF1 promoter. HEK293T cells were cotransfected with the IRF1 reporter, vector, and Flag-NLK (100 ng) expression plasmids. After 24 h, the cells were treated with IFN-γ (100 ng/ml) for 12 h prior to performing the luciferase assay. **c** NLK inhibits SeV-triggered ISRE signaling. HEK293T cells were cotransfected with the ISRE reporter and Flag-NLK (100 ng) expression plasmids. After 24 h, the cells were infected with SeV for 12 h prior to performing the luciferase assays. **d** Effects of NLK on SeV-induced endogenous *IFNB1, ISG15, ISG56,* and *RANTES* expression. HEK293T cells were transfected with Flag-NLK (1 µg) expression plasmids for 24 h followed by infection of SeV for 12 h before real-time PCR analyses were performed. **e** Immunoblot analysis of NLK and MAVS localization in mitochondria and peroxisomes under SeV stimulation. The cell fractions separated after SeV treatment were subjected to immunoblot analysis with the indicated antibodies. TOM20 was used as a mitochondrial loading control, and PMP70 was used as a peroxisome loading control. **f, g** Immunofluorescence and immunoblot analyses of NLK protein localization in mitochondria and peroxisomes under SeV stimulation. NLK, the mitochondrial marker TOM20 and the peroxisome marker Catalase were stained using the indicated method. The nuclei were stained with DAPI (blue). Images were obtained by fluorescence microscopy. Scale bar is 50 µm. E: exon. P: primer. HM: heavy mitochondria; LM: light mitochondria; Pex: peroxisome. Vec: vector plasmid. Data are representative of three independent experiments. Data are presented as the mean ± SEM ($n = 3$ for **a–c**, $n = 2$ for **d**). Statistical significance was analyzed by ANOVA ($*p < 0.05$, $**p < 0.01$, $***p < 0.001$). Source data (**a–e**) are provided as a Source Data file

cells, MCF7 breast adenocarcinoma cells, HeLa cervix adenocarcinoma cells, HepG2 liver carcinoma cells, and A549 lung carcinoma cells. Notably, NLK protein expression was not highest in TPH-1 human immune cells (Supplementary Fig. 1e). These two experiments suggested that the location and activity of NLK might change in response to SeV, but these changes are not attributable to alterations at the mRNA or protein level. NLK reportedly localizes to the Golgi apparatus[36], but whether it also localizes to mitochondria and peroxisomes, which are antiviral platforms under infection[37,38], has not been examined in previous studies. To understand whether the localization of NLK changes in response to SeV stimulation, cell fractionation and immunofluorescence (IF) experiments were performed. NLK was present in both mitochondrial and peroxisomal fractions in unstimulated cells. Interestingly, infection of cells with SeV resulted in the redistribution of NLK into mitochondrial and peroxisomal fractions (Fig. 1e–g). These data suggest that NLK is a negative regulator of SeV-induced type I interferon signaling and plays an important role in the innate immune response.

**NLK deficiency enhances the cellular antiviral response.** NLK deficiency in mice is lethal[39]. To elucidate the role of NLK in vivo, we generated conditional myeloid-deficient *Nlk* mice (*Nlk*[fl/fl/Lyz2-Cre]). Deletion of *Nlk* was verified through genomic PCR, real-time PCR and western blotting (Fig. 2a). To determine whether NLK deficiency increased the production of IFN-β in the different cell types, we isolated bone marrow-derived macrophages (BMDMs), bone marrow dendritic cells (BMDCs) and mouse lung fibroblasts (MLFs) from *Nlk*[fl/fl/Lyz2-Cre] mice and used

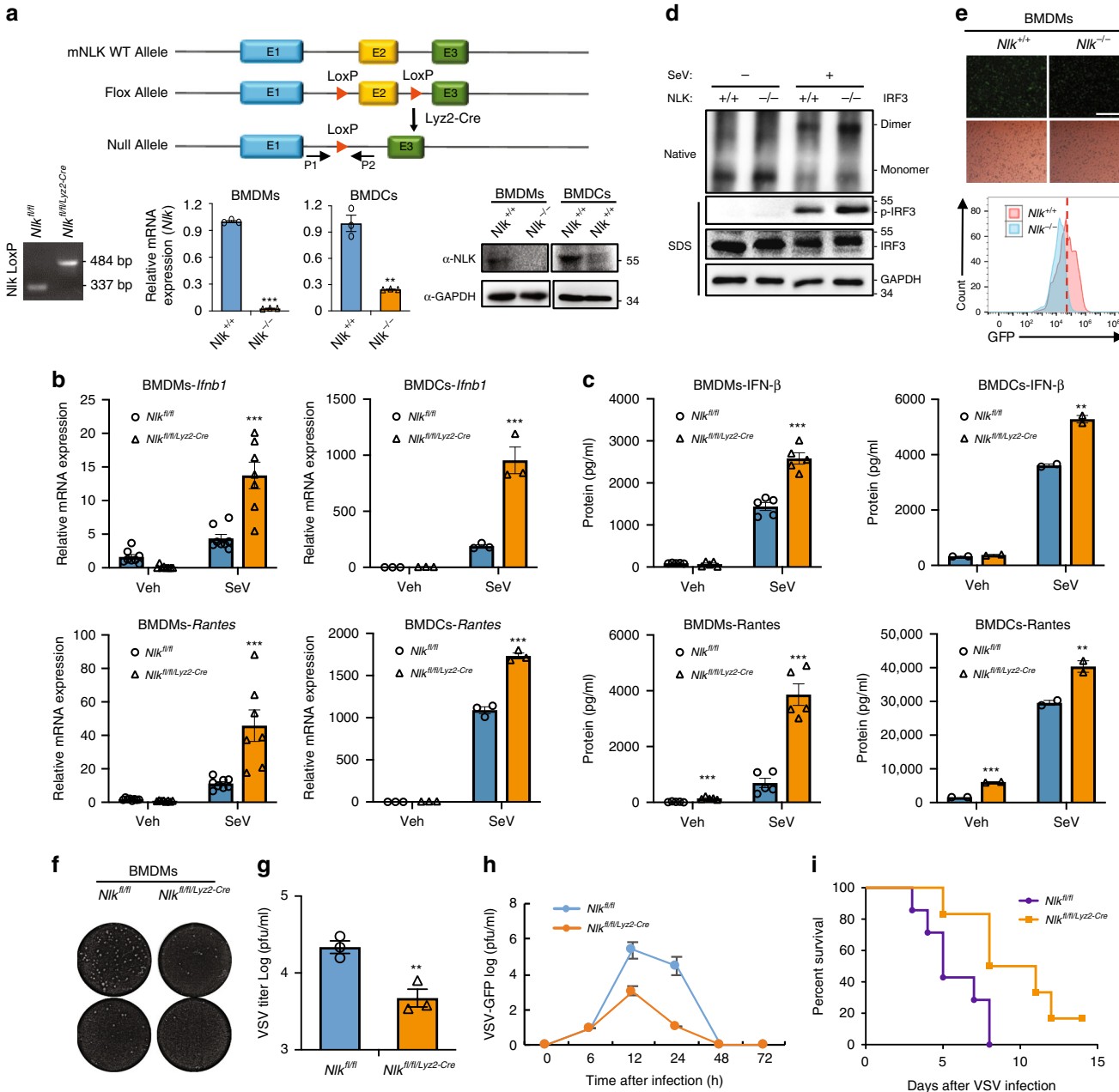

**Fig. 2** NLK deficiency enhances the SeV-induced antiviral response in mice. **a** Generation and identification of mice with a conditional myeloid deletion of *Nlk*[fl/fl/Lyz2-Cre]. Top panel, targeting strategy for the deletion of exon 3 of NLK through homologous recombination. Bottom panel, identification of NLK deletion at the genome, mRNA and protein levels. **b**, **c** NLK deficiency potentiates SeV-induced *Ifnb1* and *Rantes* gene and protein expression in BMDMs and BMDCs. NLK[+/+] and NLK[−/−] BMDMs as well as BMDCs were infected with SeV before performing real-time PCR experiments or ELISA ($n = 7$ or 3 for **b** and $n = 5$ or 2 for **c**). **d** NLK deletion facilitates SeV-induced IRF3 dimerization and phosphorylation in BMDMs. NLK[+/+] and NLK[−/−] BMDMs were infected with SeV or left untreated for 6 h. The cell lysates were then separated via native (top) or SDS (bottom three panels)-PAGE and immunoblotted with the indicated antibodies. **e** NLK deficiency in BMDMs inhibits VSV-GFP infection. NLK[+/+] and NLK[−/−] cells were infected with VSV-GFP at an MOI of 4 for 12 h before phase-contrast and fluorescence microscopy and flow cytometry analysis. Scale bar is 500 μm. **f**, **g** NLK deficiency in BMDMs potentiates the antiviral response. NLK[+/+] and NLK[−/−] cells were infected with VSV at an MOI of 4, and the culture supernatants were diluted and added to Vero cells. After fixation, cells not killed by the virus were stained with crystal violet, and the viral titer was determined ($n = 3$). **h** Wild-type and *Nlk*[fl/fl/Lyz2-Cre] mice were infected with VSV-GFP ($2 \times 10^7$ pfu) via tail vein injection. Their sera were collected at different time points and subjected to plaque assays ($n = 6$). **i** Survival assays of wild-type and *Nlk*[fl/fl/Lyz2-Cre] mice (6–7 weeks). Wild-type and *Nlk*[fl/fl/Lyz2-Cre] mice were infected with wild-type VSV ($2 \times 10^7$ pfu) via tail vein injection. The survival of the mice was monitored for 2 weeks ($n = 6$). Veh: vehicle medium. Data are representative of three **b–i** independent experiments. Data are presented as the mean ± SEM. Statistical significance was analyzed by ANOVA or Student's *t*-test (*$p < 0.05$, **$p < 0.01$, ***$p < 0.001$). Source data (**a–d**, **g–i**) are provided as a Source Data file

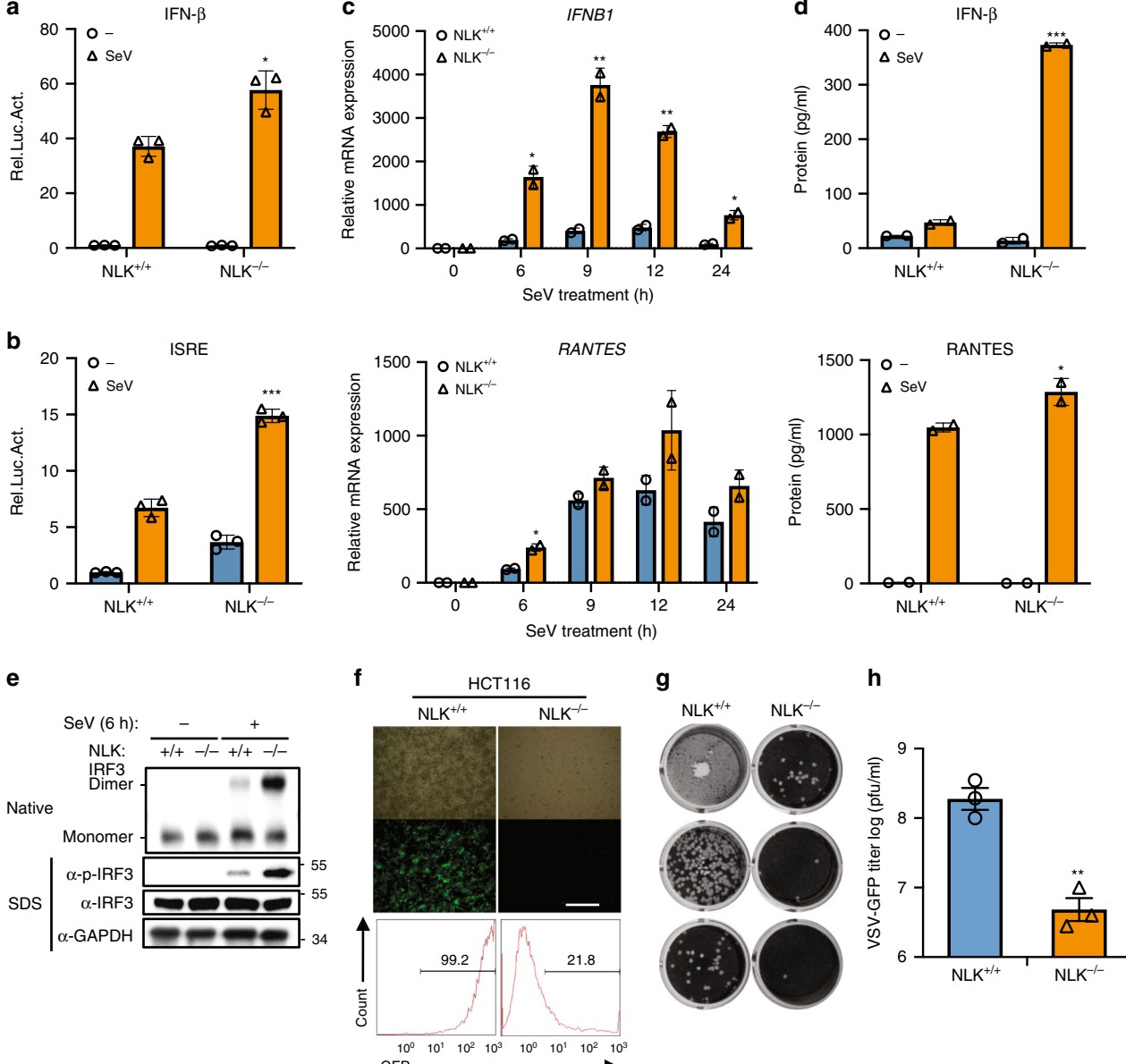

**Fig. 3** NLK deficiency promotes the cellular antiviral response in HCT116 cells. **a, b** Effects of NLK deficiency on SeV-induced IFN-β and ISRE activation. NLK$^{+/+}$ and NLK$^{-/-}$ HCT116 cells were transfected with the IFN-β or ISRE reporter plasmid (200 ng each). After 24 h of transfection, the cells were infected with SeV for 12 h before the luciferase experiments were performed ($n = 3$). **c** NLK deficiency potentiates SeV-induced endogenous *IFNB1* and *RANTES* gene expression in HCT116 cells. NLK$^{+/+}$ and NLK$^{-/-}$ cells were infected with SeV at the indicated time points before performing real-time PCR experiments ($n = 2$). **d** Effects of NLK deficiency on the SeV-induced IFN-β and RANTES protein levels. NLK$^{+/+}$ and NLK$^{-/-}$ cells were infected with SeV or left untreated for 18 h. The cell medium was then subjected to ELISA ($n = 2$). **e** NLK deficiency facilitates SeV-induced IRF3 dimerization and phosphorylation. NLK$^{+/+}$ and NLK$^{-/-}$ cells were infected with SeV or left untreated for 6 h. The cell lysates were then separated through native (top) or SDS (bottom three panels)-PAGE and immunoblotted with the indicated antibodies. **f** NLK deficiency in HCT116 cells inhibits VSV-GFP infection. NLK$^{+/+}$ and NLK$^{-/-}$ cells were infected with VSV-GFP at an MOI of 0.01 for 12 h before phase-contrast and fluorescence microscopy and flow cytometry analysis. Scale bar is 500 μm. **g, h** NLK deficiency in HCT116 cells potentiates the antiviral response. NLK$^{+/+}$ and NLK$^{-/-}$ HCT116 cells were infected with VSV at an MOI of 0.1, and the culture supernatants were diluted and added to Vero cells. After fixation, cells not killed by the virus were stained with crystal violet, and the viral titer was determined ($n = 3$). Data are representative of three independent experiments. Data are presented as the mean ± SEM. Statistical significance was analyzed by ANOVA or Student's $t$-test (*$p < 0.05$, **$p < 0.01$, ***$p < 0.001$). Source data (**a–e**, **h**) are provided as a Source Data file

previously generated HCT116/NLK$^{-/-}$ cells[35]. We assessed the effects of NLK deficiency on type I interferon activation in human cells. Using the IFN-β luciferase reporter assay, we found that NLK deficiency resulted in markedly enhanced SeV-triggered IFN-β luciferase reporter activity compared with that in the HCT116 parental cells (Fig. 3a). Similar results were obtained when NLK$^{-/-}$ cells were transfected with the ISRE luciferase

reporter plasmid following treatment with SeV (Fig. 3b). We further assessed the effects of NLK deficiency on type I interferon activation by evaluating the mRNA and protein levels of IFN-β and RANTES, revealing that NLK deficiency resulted in marked enhancement of the SeV-triggered IFN-β and RANTES mRNA and protein levels compared with those in wild-type parental cells of the different cell lines (Fig. 2b, c, Fig. 3c, d and Supplementary

Fig. 1f–h). Phosphorylation and dimerization are considered hallmarks of IRF3 activation, which in turn promote the virus-induced transcription of downstream target genes[40,41]. To determine whether NLK deficiency potentiates IRF3 activation, we examined IRF3 phosphorylation and dimerization in NLK$^{-/-}$ and wild-type cells. We found markedly enhanced SeV-induced IRF3 phosphorylation and dimer formation in NLK$^{-/-}$ cells compared with those in wild-type cells after viral infection, indicating that NLK regulates SeV-induced IRF3 signaling (Figs. 2d and 3e). To demonstrate a link between the enhanced type I interferon response and antiviral immunity in NLK$^{-/-}$ cells, we infected NLK$^{-/-}$ cells with a VSV expressing GFP (VSV-GFP) or wild-type VSV. NLK deficiency rendered the cells resistant to viral infection and resulted in a considerably lower percentage of GFP-positive cells than that in the wild-type cell population (Figs. 2e and 3f). The gating strategy shown as in the Supplementary Fig. 6. Plaque assays showed that viral infection of wild-type cells at a multiplicity of infection (MOI) of 4 in BMDMs and 0.01 in HCT116 cells led to the death of most cells within 12 h. Conversely, virus-induced cell death was prevented in NLK-deficient cells (Figs. 2f and 3g). Viral titer measurements demonstrated a potent decrease in the viral titer in NLK-deficient cells compared with that in wild-type cells (Figs. 2g and 3h). Next, we injected VSV-GFP into conditional myeloid *Nlk*-deficient and wild-type mice via the tail vein and then collected sera at the indicated times after infection to measure viral titers. The viral titers of the *Nlk*$^{fl/fl/Lyz2-Cre}$ mice were clearly lower than those of wild-type mice (Fig. 2h). To determine whether NLK deficiency could affect the survival of mice, we infected mice with wild-type VSV and monitored them for 2 weeks. As shown in Fig. 2i, all the wild-type mice died within 8 days after viral injection, but the *Nlk*-deficient mice exhibited enhanced survival after VSV injection. These data suggested that NLK deficiency markedly enhanced the type I interferon response and antiviral immunity both in vitro and in vivo.

**NLK restores the depletion-induced antiviral response.** To conveniently assay type I interferon regulation and the antiviral response, we used retroviral-mediated gene transfer to obtain stable NLK$^{-/-}$ HCT116 cells reconstituted with NLK or a kinase-dead NLK mutant NLK$^{KM}$ (Fig. 4a). Real-time PCR analyses showed that SeV infection resulted in markedly reduced *IFNB1* and *RANTES* mRNA expression levels at various time points in NLK (but not NLK$^{KM}$)-reconstituted cells compared with those in NLK$^{-/-}$ HCT116 cells (Fig. 4b). Enzyme-linked immunosorbent assays (ELISAs) of the IFN-β and RANTES protein levels yielded similar results (Fig. 4c). Antiviral response analyses using flow cytometry and plaque assays consistently demonstrated that NLK reconstitution in NLK$^{-/-}$ HCT116 cells significantly restored the frequency and viral titer of infected GFP-positive cells after VSV-GFP infection (Fig. 4d, e). By contrast, NLK$^{KM}$ reconstitution in NLK$^{-/-}$ HCT116 cells had a modest effect on the frequency and viral titer of infected GFP-positive cells compared with those of NLK$^{-/-}$ HCT116 cells maintained under the same conditions (Fig. 4d, e). NLK is a serine/threonine protein kinase, and its kinase activity is important for its function. The above results showed that NLK$^{KM}$ could not completely ameliorate the inhibition of NLK-induced IFN-β production. As IFN-β activation requires coordination between the activation of NF-κB and IRF3, we tested whether the kinase activity of NLK was essential for NF-κB and IRF3 signaling. The SeV-induced phosphorylation of IκBα was potently inhibited by NLK and NLK$^{KM}$, but phosphorylation of IRF3 was potently inhibited by NLK rather than NLK$^{KM}$, suggesting that NLK kinase activity is essential for IRF3 but not for NF-κB signaling (Fig. 4f). These

results indicate that the kinase activity of NLK is required for its roles in viral-induced IFN-β production and the antiviral response.

**NLK interacts with MAVS on mitochondria and peroxisomes.** Because NLK inhibited the viral-induced activation of type I interferon signaling, we next sought to determine the molecular mechanisms by which NLK inhibits type I interferon signaling. We cotransfected HEK293T cells with the RIG-I, MAVS, TBK1, or IRF3 expression vector together with an NLK expression vector containing an IFN-β or ISRE luciferase reporter, and NLK potently inhibited luciferase reporter activation in the presence of RIG-I, MAVS, and TBK1 but weakly inhibited IRF3 activity (Supplementary Fig. 2a, b). We previously demonstrated that NLK impacts IRF3 phosphorylation and dimerization, suggesting that NLK may inhibit type I interferon signaling by acting upstream of IRF3. Coimmunoprecipitation and immunoblot analyses confirmed that NLK interacted with MAVS but not with TBK1 and RIG-I, suggesting that NLK exerts its action on MAVS upstream of IRF3 to orchestrate the antiviral response (Fig. 5a). Endogenous coimmunoprecipitation experiments indicated that NLK was weakly associated with MAVS in unstimulated cells, and this association was clearly strengthened at 16 h after SeV stimulation (Fig. 5b). To further confirm the association between NLK and MAVS, we performed IF analysis to check their colocalization. Although most endogenous NLK was located in the nucleus, NLK in the cytosol colocalized very well with MAVS, and this colocalization was especially enhanced in MAVS prion-like aggregates after virus infection (Fig. 5c). MAVS is located on both mitochondria and peroxisomes. To explore the location of interaction between NLK and MAVS, we constructed mito-chondria- and peroxisome-localized MAVS mutants, and coim-munoprecipitation experiments showed that NLK interacted with both mitochondria- and peroxisome-localized MAVS mutants but not with the cytosol-localized MAVS mutants (Fig. 5d). Virus infection can reportedly promote the oligomerization of MAVS into a large prion-like signaling structure. To determine whether NLK is present in the prion-like MAVS complex after viral infection, we employed semidenaturing detergent agarose gel electrophoresis (SDD-AGE)[16]. Interestingly, NLK exhibited a band similar to that of MAVS following viral infection (Fig. 5e), which implies that NLK is present in the prion-like complex together with MAVS after viral infection. These data suggest that NLK interacts with MAVS on both mitochondria and peroxi-somes and exerts its function in the prion-like MAVS complex.

**NLK degrades MAVS.** TRAF3 is a vital protein for signal transduction from MAVS to TBK1 and IRF3. To understand how NLK inhibits the antiviral response, we sought to examine whether NLK decreases the stability of the MAVS/TRAF3 complex. To achieve this goal, we cotransfected NLK, MAVS and TRAF3 into HEK293T cells. Coimmunoprecipitation did not reveal a significant ability of NLK to hinder the binding of MAVS and TRAF3, but we did observe degradation of MAVS after over-expressing NLK (Supplementary Fig. 2c). Thus, we hypothesized that NLK might phosphorylate and influence the stability of MAVS, leading to the degradation of MAVS. To test this hypothesis, we overexpressed MAVS and NLK in a dose-dependent manner. As shown in Fig. 6a, NLK decreased the protein level of MAVS, and this effect was dependent on the amount of NLK. To assess whether the kinase activity of NLK was required for the degradation of MAVS, we overexpressed MAVS together with NLK or NLK$^{KM}$ and found that the kinase activity of NLK appeared to be responsible for the degradation of MAVS (Fig. 6b). To determine whether NLK regulates the degradation of

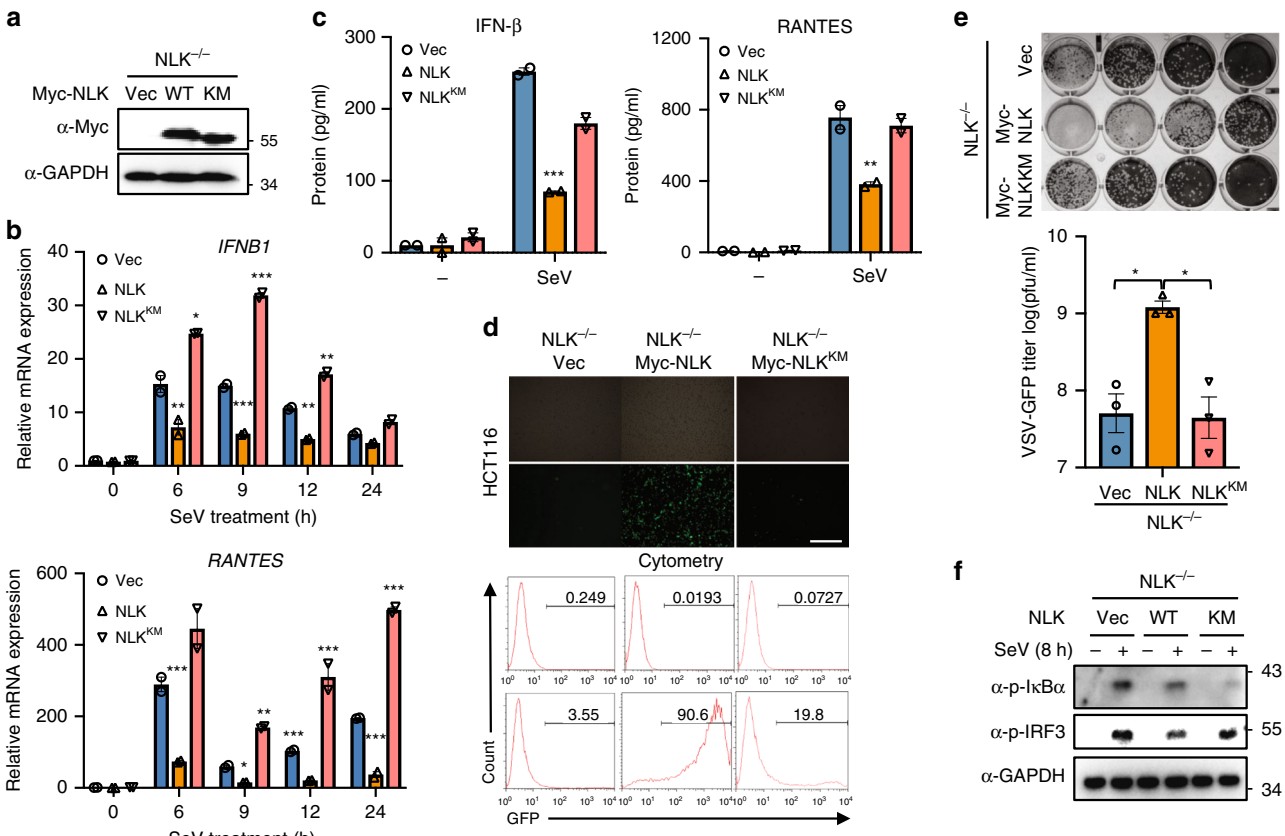

**Fig. 4** NLK, but not NLK^KM, reconstitution inhibits the cellular antiviral response. **a** The stable expression of wild-type recombinant NLK and NLK^KM in NLK^−/− HCT116 was verified. GAPDH was used as a loading control. **b** Effects of introducing the NLK or NLK^KM expression plasmid into NLK^−/− cells on SeV-induced endogenous *IFNB1* and *RANTES* gene transcription in HCT116 cells. NLK^−/− cells and NLK^−/− cells expressing the NLK or NLK^KM expression plasmids were infected with SeV at the indicated times before performing real-time PCR experiments ($n = 2$). **c** Effects of introducing the NLK or NLK^KM expression plasmid into NLK^−/− cells on SeV-induced IFN-β and RANTES protein levels. NLK^−/− cells and NLK^−/− cells expressing the NLK or NLK^KM expression plasmids were infected with SeV or left untreated for 18 h. The cell medium was subjected to ELISA ($n = 2$). **d** Effects of introducing the NLK or NLK^KM expression plasmids into NLK^−/− cells on VSV-GFP infection. NLK^−/− and NLK^−/− cells expressing the NLK or NLK^KM plasmids were infected with VSV-GFP at an MOI of 1 for 12 h before phase-contrast and fluorescence microscopy and flow cytometry analysis. Scale bar is 500 μm. **e** Effects of introducing the NLK or NLK^KM expression plasmid into NLK^−/− cells on the antiviral response. NLK^−/− and NLK^−/− cells expressing the NLK or NLK^KM plasmid were infected with VSV-GFP at an MOI of 1, and the culture supernatants were added to Vero cells. After fixation, cells not killed by the virus were stained with crystal violet, and the viral titer was determined ($n = 3$). **f** Effects of NLK or NLK^KM on the SeV-induced phosphorylation of IκBα and IRF3. NLK^−/− cells and NLK^−/− cells expressing the NLK or NLK^KM expression plasmid were infected with SeV or left untreated for 8 h. Immunoblotting was performed using the indicated antibodies. GAPDH was used as a loading control. NLK^KM: Serine 167 to Alanine. Data are representative of three independent experiments. Data are presented as the mean ± SEM. Statistical significance was analyzed by ANOVA (*$p < 0.05$, **$p < 0.01$, ***$p < 0.001$). Source data (**a–c**, **e–f**) are provided as a Source Data file

endogenous MAVS, we performed a similar experiment, and the result was consistent with that shown in Fig. 6b (Fig. 6c). To further assess the influence of NLK on the stability of MAVS, we used a cycloheximide-chase (CHX) assay based on the time course of MAVS degradation. An increase in the half-life of MAVS in NLK-depleted BMDMs and HCT116 cells supported the hypothesis that NLK stabilizes the protein expression of MAVS (Fig. 6d and Supplementary Fig. 2d). Stimulation of the virus-induced degradation of MAVS using SeV induced the degradation of MAVS in NLK-depleted BMDMs, which suggested that NLK regulated the SeV-induced degradation of MAVS (Fig. 6e). These data suggest that NLK mediates the destabilization of MAVS.

**NLK phosphorylation of MAVS switches the degradation**. We noted that NLK not only reduced the protein level of MAVS but also resulted in its shifting, as shown in Fig. 5a. Given that NLK is a serine/threonine protein kinase, we believe that MAVS

undergoes phosphorylation. To verify this hypothesis, we carried out a ³²P labeling kinase assay in vitro. We failed to express GST-NLK and MAVS in bacteria; instead, Flag-NLK, Flag-NLK^KM, and HA-MAVS were purified from the HEK293T cells using a high salt concentration to wash out protein impurities. These data indicated that NLK, but not the NLK^KM mutant, directly phosphorylated MAVS (Fig. 7a). This result was further supported using a phospho-(Ser/Thr) antibody (Supplementary Fig. 3a), which also suggested that the shift in MAVS required the kinase activity of NLK. To further confirm that this PTM-induced shift was caused by phosphorylation, we sought to determine whether λ-PPase could prohibit this modification. As shown in Fig. 7b, λ-PPase inhibited the MAVS shift in the presence of NLK. Next, we investigated the residues of MAVS that were phosphorylated by NLK. Masspec was employed to analyze the PTM of MAVS in the presence or absence of NLK or NLK^KM. According to the obtained data, MAVS is subject to housekeeper phosphorylation events at serines 152/165/222 in the presence of NLK as well as at

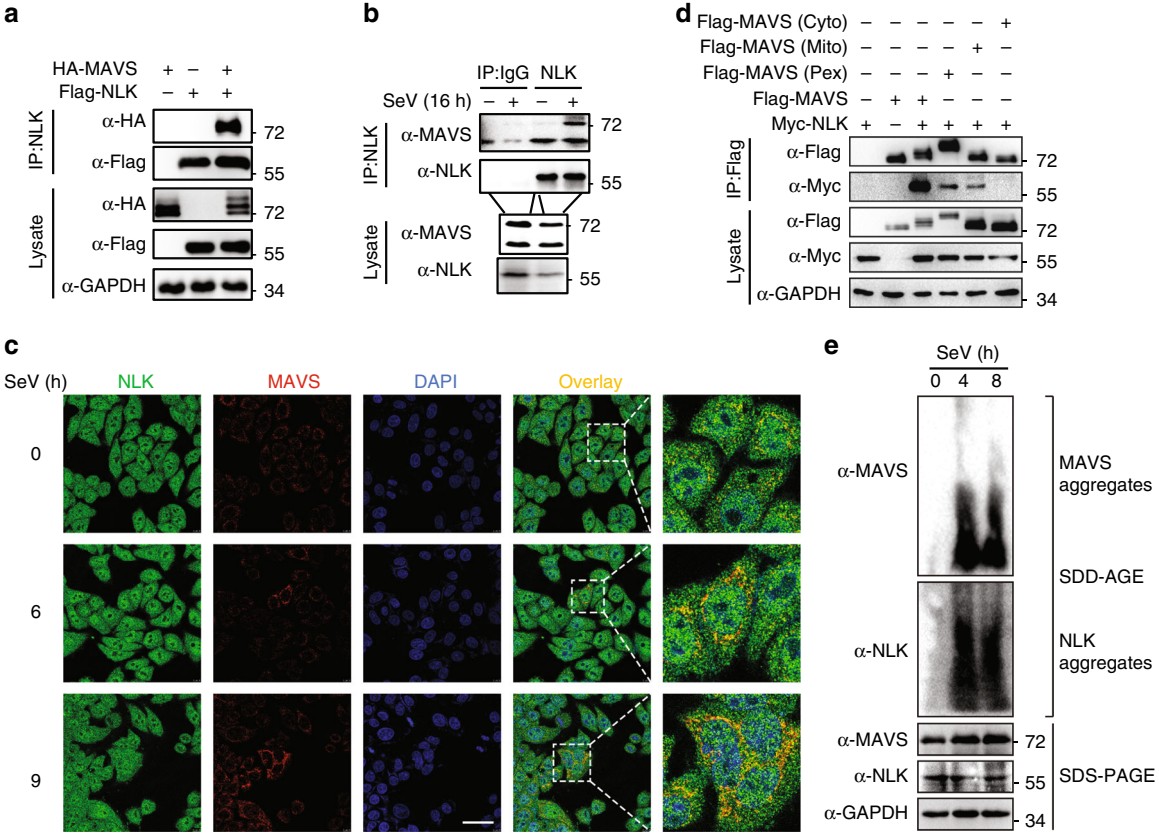

**Fig. 5** NLK interacts with MAVS on both mitochondria and peroxisomes. **a**, **b** Coimmunoprecipitation analysis of the interaction between NLK and MAVS. **a** HEK293T cells were transfected with HA-MAVS and Flag-NLK. Coimmunoprecipitation and immunoblot analyses were performed with the indicated antibodies. **b** Endogenous coimmunoprecipitation was performed using HEK293T cells in the presence or absence of SeV stimulation. The immunoblot analysis was performed with the indicated antibodies. **c** Immunofluorescence analysis of the colocalization of endogenous NLK and MAVS after SeV infection. NLK (green) and MAVS (red) were stained using the indicated antibodies. The nuclei were stained with DAPI (blue). Images were obtained by fluorescence microscopy. Scale bar is 50 μm. **d** Coimmunoprecipitation analysis of the interaction between NLK and the MAVS mutant. HEK293T cells were transfected with wild-type-, cytosol- (cyto), mitochondria- (mito) or peroxisome (pex)-localized Flag-MAVS as well as with Myc-NLK. Coimmunoprecipitation and immunoblot analyses were performed with the indicated antibodies. **e** HEK293T cells were infected with SeV for the indicated time, and aliquots of the extracts were then analyzed by SDD-AGE or SDS-PAGE. Cyto: cytosol location. Mito: mitochondria location. Pex: peroxisomes location. Data are representative of three (**a**, **b**, **e**) or two (**c**, **d**) independent experiments. Source data, **a**, **b**, **d**, **e**, are provided as a Source Data file

4 other sites, serines 121/212/258/329 (Fig. 7c). To identify the most important site for MAVS function, we constructed a MAVS mutant by changing these 4 serines to alanines, either individually (MAVS[S121A], MAVS[S212A], MAVS[S258A], and MAVS[S329A]) or all together (MAVS[S4A]). MAVS[S4A] possessed the strongest ability to induce *IFNβ* production in the presence of NLK (Supplementary Fig. 3b); therefore, we used this mutant to study antiviral function. To determine whether the phosphorylation of the 4 residues plays a role in antiviral activity, we used the MAVS mutant MAVS[S4A] to perform plaque assays. MAVS[S4A] promoted antiviral effects in the presence of NLK and in MAVS[−/−] HEK293T cells compared with that in wild-type MAVS HEK293T cells (Fig. 7d). We explored whether MAVS[S4A] could block the shift induced by NLK, revealing that MAVS[S4A] completely abrogated this shift when MAVS and MAVS[S4A] were expressed in the presence of NLK (Fig. 7e). Next, we asked whether MAVS[S4A] could also alter the stability of MAVS in MAVS[−/−] HEK293T cells. To address this question, the CHX assay was performed, which showed that MAVS[S4A] displayed a longer half-life than wild-type MAVS (Fig. 7f). To determine whether the SeV-induced degradation of MAVS is affected by MAVS[S4A], we expressed MAVS and MAVS[S4A] in MAVS[−/−] cells and stimulated them with SeV to induce MAVS degradation. As shown in Fig. 7g, MAVS[S4A] blocked SeV-induced degradation,

which indicated that SeV-induced MAVS degradation requires the phosphorylation of MAVS at the 4 serine residues. To determine whether the antiviral effects of NLK depend on MAVS, we employed DNA ligand-induced non-MAVS-dependent activities in NLK[−/−] BMDMs. NLK did not affect the mRNA levels of *Ifnb1* and *Rantes* (Supplementary Fig. 3c, d). Taken together, these data suggested that NLK phosphorylates MAVS at 4 serine residues to switch the subsequent degradation induced by SeV.

**MAVS-derived peptide promotes the antiviral response.** Experiments performed to narrow down the MAVS region responsible for its interaction with NLK revealed the C-terminal region from aa451-aa500 to be the relevant region (Supplementary Fig. 4a, b). The transmembrane domain (TM) was shown to be required for the mitochondria or peroxisome localization of MAVS, and loss of this region resulted in the disaggregation of MAVS and NLK (Supplementary Fig. 4b), which indicated that the interaction occurred on mitochondria or peroxisome. Further mapping of the aa451-aa500 region of MAVS required for NLK binding revealed that the aa471-aa480 peptide (denoted by NLK-associated and MAVS-derived antiviral peptide (NAMDAP) herein) was necessary for the interaction between MAVS and NLK (Fig. 8a). First, we assessed the competitive role of

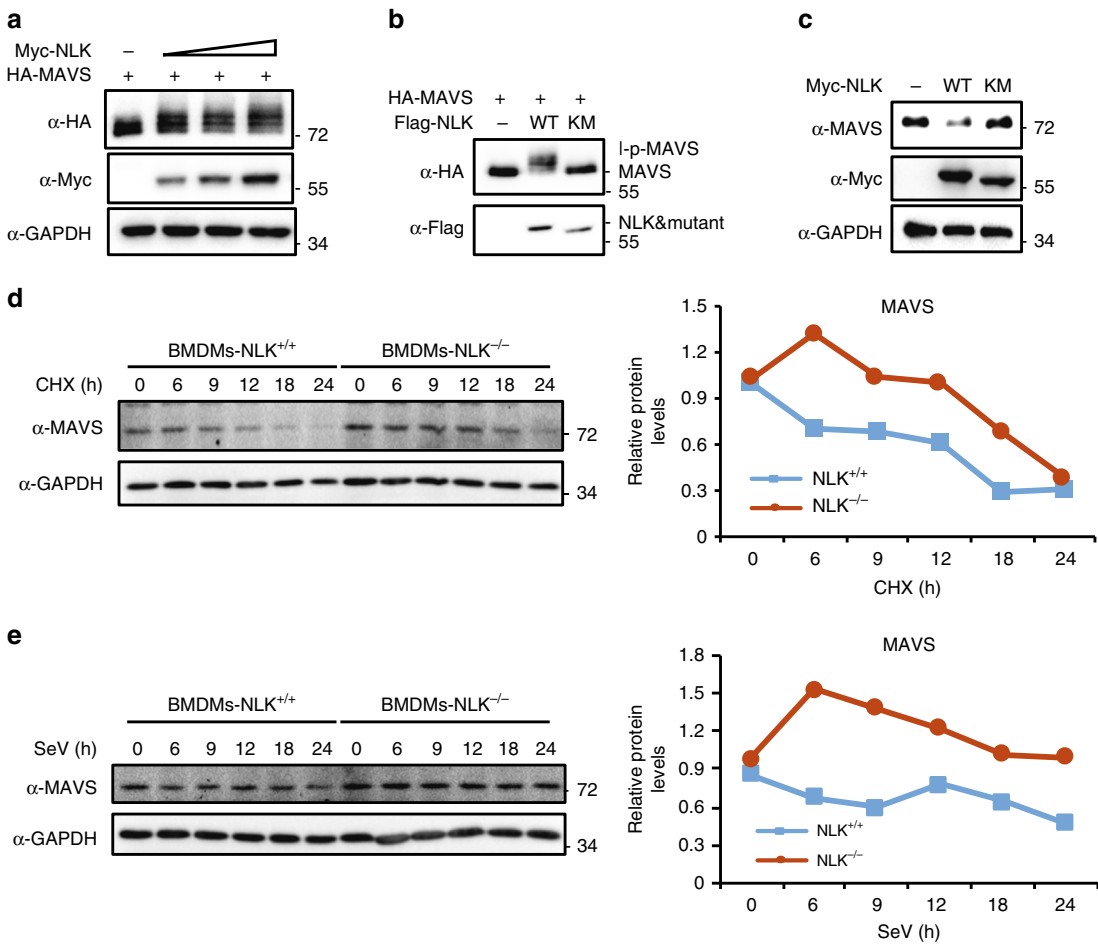

**Fig. 6** NLK mediates the degradation of MAVS. **a** NLK-induced dose-dependent degradation of MAVS. HEK293T cells were transfected with HA-MAVS and increasing amounts of Myc-NLK. The protein levels of MAVS and NLK were determined after transfection for 12 h through immunoblotting using the HA and Myc antibodies. **b**, **c** NLK and NLK$^{KM}$ were analyzed for MAVS degradation. **b** HEK293T cells were transfected with HA-MAVS and Flag-NLK or Flag-NLK$^{KM}$. The protein levels of MAVS and NLK were determined via immunoblot analysis using HA and Flag antibodies. **c** HEK293T cells were transfected with Myc-NLK or Myc-NLK$^{KM}$. The protein levels of MAVS and NLK were determined through immunoblot analysis using MAVS and Myc antibodies. **d** Stability of MAVS in wild-type and NLK-deficient BMDMs. BMDMs were treated with CHX for the indicated periods, and the cell lysate was then subjected to immunoblot analysis using the MAVS antibody. Relative levels of MAVS were calculated after normalization to GAPDH and are shown in the right panel. **e** Stability of MAVS in wild-type and NLK-deficient BMDMs after SeV infection. The BMDMs were treated with CHX for the indicated periods, and the cell lysate was then subjected to immunoblot analysis using MAVS antibodies. The relative levels of MAVS were calculated after normalization to GAPDH and are shown in the right panel. Data are representative of three independent experiments. Source data (**a**–**e**) are provided as a Source Data file

NAMDAP for binding between MAVS and NLK, revealing that NAMDAP effectively reduced the association between MAVS and NLK (Fig. 8b). Thus, we also determined whether NAMDAP protects MAVS from degradation induced by the interaction between NLK and MAVS. Upon the addition of NAMDAP at different doses, the MAVS shift was ameliorated in the presence of 10 μg/ml NAMDAP, and MAVS became more abundant with increasing amounts of NAMDAP (Supplementary Fig. 4c). Then, we assessed whether NAMDAP could rescue the inhibitory effect of NLK on MAVS-induced IFN production and the antiviral response. By analyzing the mRNA levels of IFN and viral replication, NAMDAP was found to reverse the effect of NLK on MAVS (Supplementary Fig. 4d, e). Moreover, we performed a similar experiment to confirm whether NAMDAP could influence the effect of NLK on endogenous MAVS. Not surprisingly, the results were consistent with those shown in Supplementary Fig. 4d, e (Fig. 8c, d), suggesting that NAMDAP propagated the NLK-induced inhibition of the antiviral response. Next, we examined whether NAMDAP could potentiate the antiviral

response after viral infection. To achieve this goal, we assessed the antiviral response following the addition of NAMDAP after VSV-GFP infection using flow cytometry and plaque assays (Fig. 8e, f). As shown in the figures, NAMDAP promoted the VSV-GFP-induced antiviral response. Additionally, NAMDAP increased the mRNA expression levels of the antiviral cytokines *IFN-β* and *RANTES* (Fig. 8g and Supplementary Fig. 4f). To further verify the antiviral effects of NAMDAP in vivo, mice were infected with VSV-GFP and NAMDAP via tail vein injection. At 12 h after VSV-GFP and NAMDAP injection, sera were collected to measure viral titers. The titers were reduced in mice injected with NAMDAP compared with those in mice not receiving the NAMDAP injection (Fig. 8h). We continued to observe the survival of mice after VSV-GFP and NAMDAP injection for 14 days. Surprisingly, injecting mice with NAMDAP substantially increased their survival, whereas all of the mice not injected with NAMDAP died before day 7 (Fig. 8i). Taken together, these results suggest that NAMDAP could be a potent antiviral drug for attenuating viral infection.

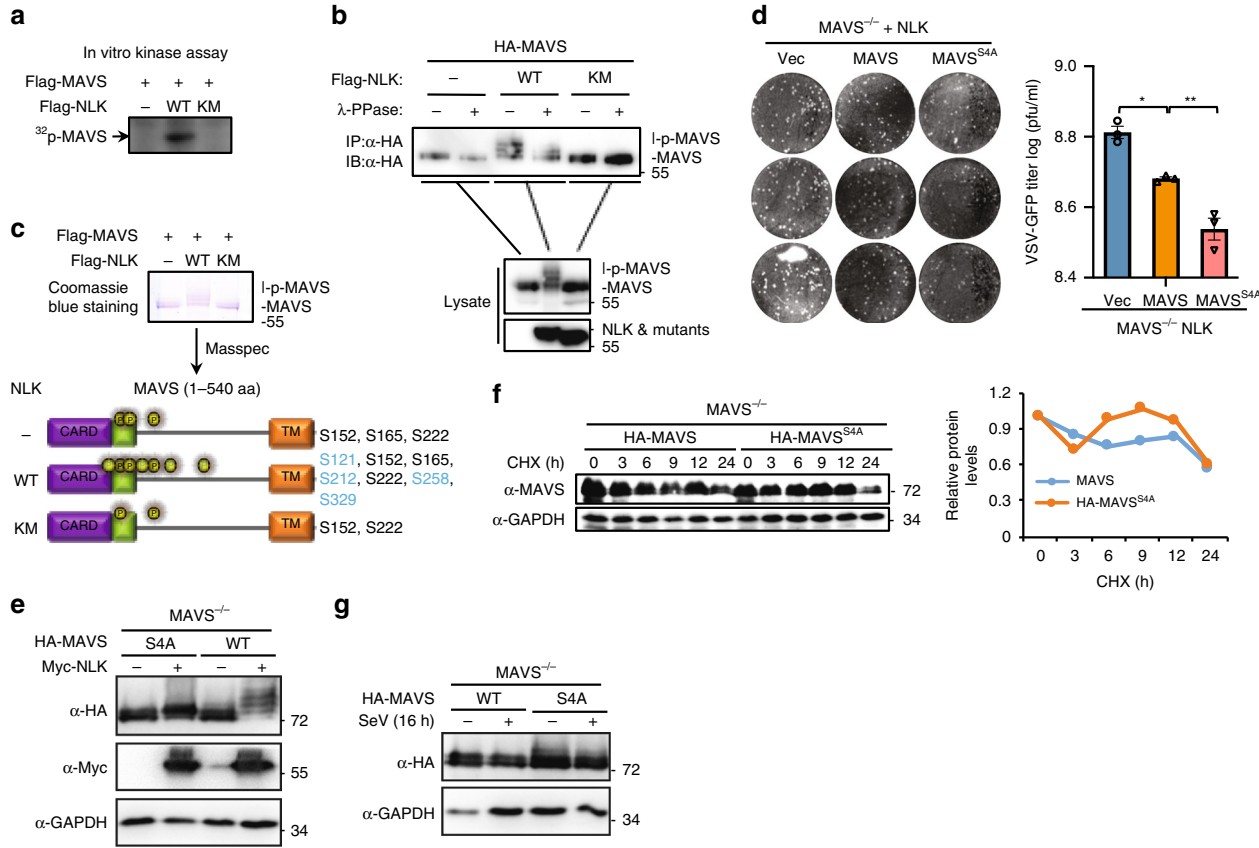

**Fig. 7** NLK phosphorylates MAVS and alters proteasome-dependent degradation. **a** Phosphorylation of the MAVS protein by NLK in vitro. HEK293T cells were transfected with the indicated expression vectors. The protein immunoprecipitated with the anti-Flag antibody was incubated with [γ-$^{32}$P] ATP. The phosphorylated proteins were visualized through autoradiography. **b** The NLK-dependent mobility shift of MAVS due to phosphorylation. The proteins immunoprecipitated with the anti-HA antibody were incubated with γ-PPase. The samples were analyzed via immunoblot using the indicated antibodies. **c** Phosphorylation sites of MAVS as detected by Masspec in the absence or presence of NLK or NLK$^{KM}$. The proteins immunoprecipitated with the anti-Flag antibody were subjected to SDS-PAGE and stained with Coomassie blue following mass spectrometry analysis. The phosphorylation sites are shown in the graphs. The highlighted sites are specific for the presence of NLK (bottom panel). **d** Effects of MAVS or MAVS$^{S4A}$ in MAVS$^{-/-}$ HEK293T cells expressing NLK on the antiviral response. MAVS$^{-/-}$ HEK293T cells were transfected with NLK and MAVS or MAVS$^{S4A}$ and then infected with VSV-GFP at an MOI of 1, and the supernatants were added to Vero cells. After fixation, cells not killed by the virus were stained with crystal violet (left panel), and the viral titer was then determined (right panel) ($n=3$). **e** Effects of NLK on MAVS and MAVS$^{S4A}$. MAVS$^{-/-}$ HEK293T cells were transfected with the indicated plasmids, and cell lysates were analyzed via immunoblot. **f** Stability of MAVS and MAVS$^{S4A}$ in MAVS$^{-/-}$ HEK293T cells. The cells were treated with CHX for the indicated periods and then subjected to immunoblot analysis. The relative levels of MAVS were calculated after normalization to GAPDH. **g** Effect of SeV-induced MAVS degradation on wild-type HA-MAVS and HA-MAVS$^{S4A}$. MAVS$^{-/-}$ HEK293T cells were transfected with HA-MAVS and MAVS$^{S4A}$, after 12 h, incubation with SeV for another 16 h. The cell lysate was then analyzed via immunoblot using the HA antibody. MAVS$^{S4A}$: Serine 121/212/258/329 to Alanine. Data are representative of three independent experiments. Data are presented as the mean ± SEM. Statistical significance was analyzed by ANOVA or Student's $t$-test (*$p < 0.05$, ***$p < 0.001$). Source data **a**–**g** are provided as a Source Data file

## Discussion

RLRs detect viruses to mediate the induction of genes that encode inflammatory cytokines, interferons and interferon-related molecules. In general, RLRs that recognize viruses induce proinflammatory cytokines and interferons to trigger interferon and innate immune responses[42,43]. As a mitochondrial antiviral adapter protein, MAVS is vital for the RLR-induced signaling cascade[44]. To maintain antiviral innate immune homeostasis, MAVS is degraded during the later phase after virus infection[45]. The Trim25 and AIP4 ubiquitin E3 ligases have been reported to mediate the viral-triggered ubiquitination and degradation of MAVS,[17,18,46] however, the kinase that regulates MAVS phosphorylation and results in its degradation to maintain innate immune homeostasis has remained elusive. In this study, we identified a role of NLK in the negative regulation of type I interferon signaling and demonstrated the molecular mechanisms by which NLK

regulates MAVS phosphorylation and degradation and influences its signaling through phosphorylation at multiple serine sites. We also identified an antiviral peptide, NAMDAP, that displayed obvious antiviral effects in vitro and in vivo.

As a type of PTM, phosphorylation plays a vital role in signaling, including that in the innate immune system. However, it is not clear whether phosphorylation is involved in the regulation of MAVS. Therefore, we screened more than 100 kinases to identify potential genes that play important roles in the innate immune system. Fortunately, one evolutionarily conserved serine/threonine protein kinase, NLK, was found to exert strong inhibitory effects on the production of antiviral cytokines after the infection of cells with SeV. We further identified this antiviral effect through gene overexpression or knock out. Moreover, a mouse experiment revealed a strong role of NLK in the antiviral response. We explored the mechanism by which NLK influences antiviral innate immunity. MAVS was identified as a substrate of

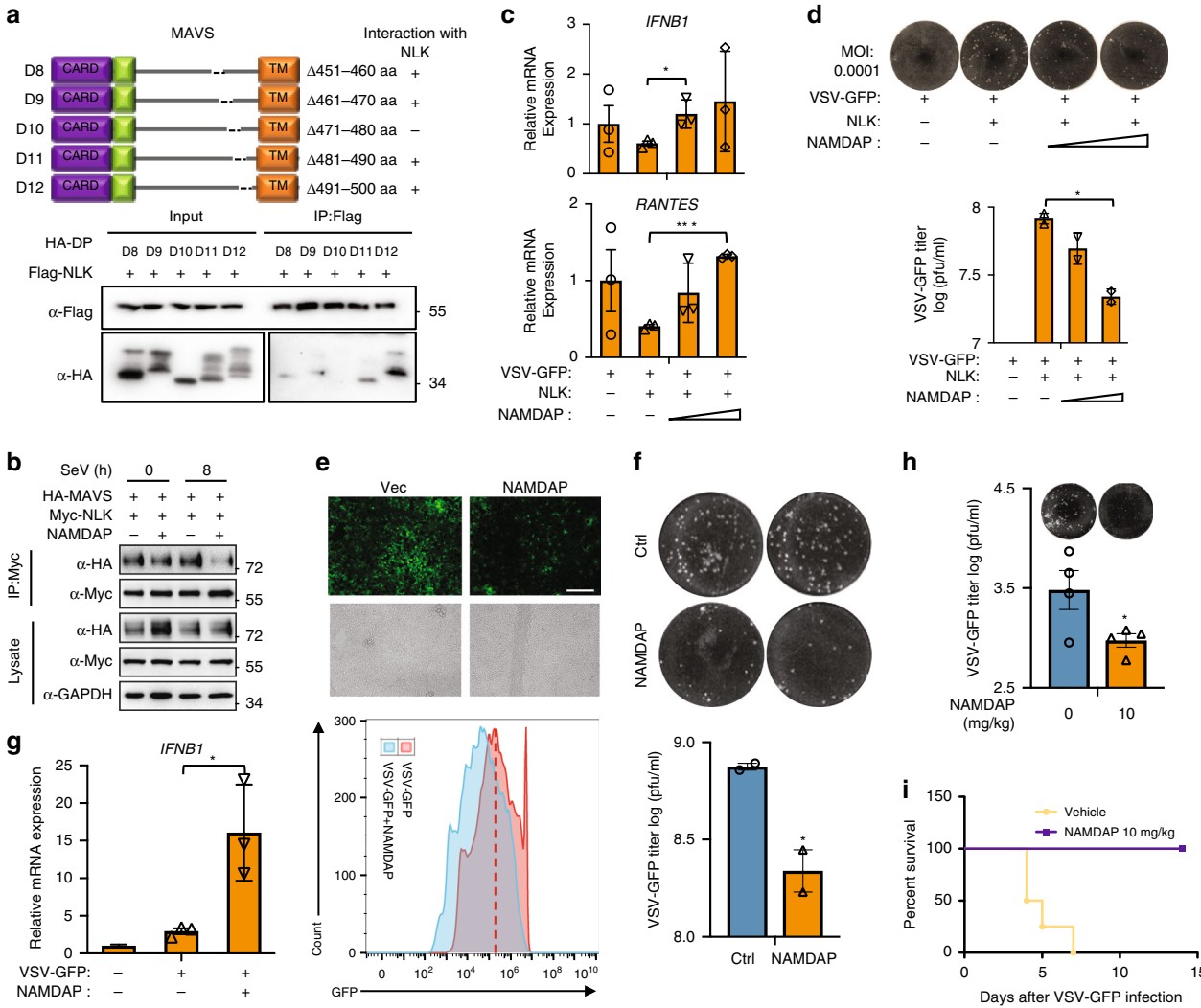

**Fig. 8** A peptide from MAVS potentiates the antiviral response in vitro and in vivo. **a** Characterization of the interaction domain between NLK and MAVS. HEK293T cells were transfected with the MAVS mutants and NLK, immunoprecipitated using the Flag antibody, and then subjected to immunoblot analysis. **b** Effect of NAMDAP on the interaction between NLK and MAVS. HEK293T cells were transfected with NLK and MAVS, MG132 was added to protect MAVS from degradation, and 20 μg/ml NAMDAP and SeV were added for 8 h before performing immunoprecipitation experiments. Immunoblot analysis was performed using the indicated antibody. **c, d** Effect of NAMDAP on the inhibitory effect of NLK on MAVS toward *IFNB1* and *RANTES* gene transcription and the antiviral response. **c** HEK293T cells expressing NLK were incubated with increasing amounts of NAMDAP before performing real-time PCR experiments ($n = 3$). **d** HEK293T cells expressing NLK were incubated with VSV-GFP at an MOI of 0.0001 and increasing amounts of NAMDAP, followed by plaque assays ($n = 2$). **e–g** Effect of NAMDAP on the production of antiviral cytokines and the antiviral response in vitro. HEK293T cells were infected with VSV-GFP at an MOI of 0.01 followed by monitoring via FACS analysis (**e**). Scale bar is 500 μm. The supernatant was collected and added to Vero cells before performing the plaque assay (**f**) ($n = 2$). mRNA was isolated to determine the transcription level of *IFNB1* (**g**) ($n = 3$). **h, i** Effect of NAMDAP on the antiviral response in vivo. Mice (12) were infected with VSV-GFP ($2 \times 10^8$ pfu), after which 10 mg/kg NAMDAP or vehicle was injected three times every 4 h via the tail vein. Their sera were collected after 12 h and subjected to the plaque assay (**g**). Survival assay of mice (four for each) injected with or without NAMDAP. The survival of the mice was monitored for 2 weeks (**h**) ($n = 4$). NAMDAP: NLK-associated and MAVS-derived antiviral peptide. Data are representative of three independent experiments. Data are presented as the mean ± SEM. Statistical significance was analyzed by ANOVA or Student's $t$-test (*$p < 0.05$, ***$p < 0.001$). Source data (**a–d**, **f–h**) are provided as a Source Data file

NLK, and phosphorylation of MAVS by NLK resulted in its degradation. The MAVS$^{S4A}$ mutant was shown to be more stable than wild-type MAVS under resting conditions and SeV-induced degradation, suggesting that the phosphorylation of MAVS by NLK switches the subsequent ubiquitination and degradation of MAVS. The effective antiviral response to the peptide revealed that domain mapping is an effective and practicable method for elucidating how to ameliorate innate immunity inhibition and thus enhance the antiviral response. The schematic diagram of NLK for innate immunity inhibition as shown in Supplementary Fig. 5.

In response to SeV stimulation, the NLK mRNA and protein levels differed slightly at various time points, indicating that NLK kinase activity or localization might change after viral stimulation. Although endogenous NLK localizes to the Golgi apparatus in the absence of stimulation[36], the isolation of cell fractions showed that NLK translocated into mitochondria or peroxisomes, where MAVS exerts its effect after viral stimulation. This observation suggested a potential association between NLK and MAVS in the antiviral response. Immunoprecipitation showed that NLK has a weak association with MAVS in resting cells. In contrast, NLK bound strongly to MAVS after exposure to the virus in the

later phase of infection. This interaction was shown to occur in both mitochondria and peroxisomes as well as to require the transmembrane domain of MAVS, further revealing the significance of mitochondria or peroxisomes as a platform in the antiviral response and showing that this interaction is necessary for the phosphorylation of MAVS by NLK. More interestingly, NLK was present in a complex with oligomeric MAVS, supporting that NLK plays an important role in antiviral response homeostasis.

The region of MAVS that interacts with NLK overlays its region that interacts with TRAF3, but the interaction between TRAF3 and MAVS is not affected by NLK. In fact, in addition to its association with MAVS, TRAF3 was also found to bind to NLK, but we did not observe an obvious change in TRAF3 protein expression, and NLK did not exert its antiviral effect through TRAF3.

We have previously shown that NLK plays an important role in TNFα-induced NF-κB signaling. We elucidated the contribution of NLK to the IRF3 and NF-κB pathways, in which the kinase activity of NLK is required for IRF3 signaling but is not necessary for the NF-κB pathway. NAMDAP plays a notable rescue role in antiviral response inhibition by NLK but a relatively weak role in promoting the virus-induced antiviral response, revealing another potent target of NLK in the innate immune response. Furthermore, NLK has been reported to suppress the expression of a wide range of genes through the CREB binding protein (CBP) coactivator[47]. Thus, NLK potentially controls the innate immunity-related transcription factor complex to negatively regulate type I interferon signaling. Our future efforts will be directed at understanding the function of NLK in this pathway. The short life span of NAMDAP should be considered to increase its antiviral effect in vitro and in vivo. Collectively, our data clearly advance our comprehension of the complicated networks and homeostasis of innate antiviral immunity.

## Methods

**Reagents and constructs**. NLK (A400-046A, 1:500) and MAVS (A300-782A, 1:1000) (Bethyl, TX, USA); NLK (monoclonal antibody produced by ourselves for IF); Flag (F1804, 1:5000) and Catalase (219010, 1:5000) (Sigma-Aldrich, St. Louis, MO, USA); Myc (CW0299M, 1:5000), GAPDH (CW0101M, 1:10000), and Tubulin (CW0098M, 1:10000) (CWBIO, Beijing, China), HA (51064-2-AP, 1:5000), COX IV (11242-1-AP, 1:2000), and TOM20 (11802-1-AP, 1:2000 for WB or 1:100 for IF) (Proteintech, Hubei, China); p-IκBα (9246, 1:1000) and p-IRF3 (29047, 1:1000) (Cell Signaling, Danvers, MA, USA); and MAVS (sc-365333 for IF, 1:100) and IRF3 (sc-376455, 1:1000) (Santa Cruz Biotechnology, Santa Cruz, CA, USA) were purchased from the indicated companies. The following reagents obtained from the indicated suppliers were used in this study: CHX (Sigma) and λ-PPase (New England Biolabs, Hitchin, UK). The IFN-β and ISRE luciferase reporter plasmids and SeV, VSV and VSV expressing GFP (VSV-GFP) were gifts from Hongbing Shu and Bo Zhong[48,49] (Wuhan University, Hubei, China). Plasmids encoding MAVS, NLK and the mutants were constructed using standard molecular cloning techniques. The sequences for DNA ligands are listed below:

ISD45: 5′-TACAGATCTACTAGTGATCTATGACTGATCTGTACATGATCT ACA-3′; HSV60: 5′-TAAGACACGATGCGATAAAATCTGTTTGTAAAATTTA TTAAGGGTACAAATTGCCCTAGC-3′; DNA90: 5′-TACAGATCTACTAGTGA TCTATGACTGATCTGTACATGATCTACATACAGATCTACTAGTGATCTAT GACTGATCTGTACATGATCTACA-3′; HSV120: 5′-AGACGGTATATTTT

TGCGTTATCACTGTCCCGGATTGGACACGGTCTTGTGGGATAGGCAT GCCCAGAAGGCATATTGGGTT AACCCCTTTTTATTTGT GGCGGGTTTTT TGGAGGACTT-3′;

**Cell lines, retroviral gene transfer and mice**. All the cell lines used in the study were originally derived from ATCC. NLK[−/−] HCT116 cells were previously generated via conventional gene knockout procedures[34,35]. Vero cells were a gift from Fenyong Liu (Wuhan University, Hubei, China). MAVS[−/−] HEK293T cells were a gift from Hongbing Shu (Wuhan University, Hubei, China). The reintroduction of wild-type or mutant NLK into NLK[−/−] HCT116 cells was performed through retroviral-mediated gene transfer. Briefly, HEK293T cells plated in 35-mm dishes were transfected with the indicated retroviral expression plasmid (2 μg) together with the pGag-pol (1 μg) and pVSV-G (1 μg) plasmids. Two days after transfection, the viruses were harvested and used to infect the indicated cells in the presence of polybrene (8 μg/mL). The infected cells were selected via puromycin (1 μg/mL)

treatment for 7 days. To isolate BMDMs or BMDCs, bone marrow cells were collected from the femurs of Nlk[fl/fl/Lyz2-Cre] mice, and blood cells were simultaneously lysed in 1XACK buffer. Then, the bone marrow cells were cultured in DMEM containing 10% FBS and M-CSF (10 ng/ml) for BMDMs or GM-CSF (20 ng/ml) for BMDCs. Three days later, the supernatant was removed, and the cells were harvested via trypsin digestion and cultured in 6-well plates for further experiments. MLFs were isolated from 10-week-old mice. Lungs were cut up and digested in HBSS buffer with 10 mg/ml type II collagenase and 20 μg/ml DNase I for 4 h at 37 °C. Cell suspensions were cultured in DMEM containing 10% FBS. The C57BL/6 NLK conditional knockout mice and Lyz2-Cre were obtained from Jackson Lab (B6.129P2-Lyz2[tm1(cre)Ifo]/J, stock no: 004781)[48]. Male or female C57BL/6 mice aged 6–8 weeks were used for the experiments. All mice were housed under a 12:12-hour light/dark cycle at a controlled temperature. All animal studies were conducted in accordance with the Guidelines of the China Animal Welfare Legislation and approved by the Committee on Ethics in the Care and Use of Laboratory Animals of Wuhan University (permit number: 15060 A). All efforts were made to minimize suffering.

**Flow cytometry**. After 16 h of VSV-GFP infection, the indicated cells were harvested and resuspended in 0.5 mL of phosphate-buffered saline (PBS). The samples were analyzed by flow cytometry (Beckman Coulter, Fullerton, CA, USA), and the flow cytometry data were analyzed using FlowJo software (TreeStar, Ashland, OR, USA).

**VSV plaque assays**. The indicated cells ($2 \times 10^5$) were infected with VSV. At 1 h after infection, the cells were washed three times with PBS, and fresh medium was then added. The supernatants were harvested at the indicated time points after washing. Thereafter, the supernatants were diluted and used to infect confluent Vero cells cultured in 12-well plates. At 1 hour post infection, the supernatant was removed, and the samples were overlaid with 1% methylcellulose. At day 2 post infection, the overlay was removed, and the cells were fixed with 4% formaldehyde for 30 min and stained with 0.25% crystal violet in 20% methanol. Finally, the plaques were counted, averaged, and multiplied by the dilution factor to determine the viral titer as PFU/mL.

**Transfection and reporter assays**. HEK293T cells ($1 \times 10^5$) were seeded in 24-well plates and transfected with plasmids encoding the indicated luciferase reporters, pRL-TK Renilla luciferase, and different expression or control vectors using the TurboFect reagent (Thermo Scientific, Waltham, MA, USA). After 24 h, a dual specific luciferase assay kit (Promega, Madison, WI, USA) was used for the reporter assays.

**Coimmunoprecipitation and immunoblot analyses**. HEK293T cells ($1 \times 10^6$) were transfected or stimulated and harvested in 400 μL of NP40 lysis buffer (30 mM Tris-HCl pH 7.4, 150 mM NaCl, and 1% NP40) with a protease inhibitor cocktail (Roche, Basel, Switzerland). Next, the supernatants were incubated with the indicated antibodies and Protein G beads (Roche) at 4 °C for 5 h. The beads were then washed three times with lysis buffer, and immunoprecipitants were eluted with SDS loading buffer and fractionated on SDS-PAGE gels. Proteins were transferred to PVDF membranes (Bio-Rad, Hercules, CA, USA) and further incubated with the indicated antibodies. The Clarity™ Western ECL Substrate System (Bio-Rad) was used for protein detection. Semidenaturing detergent agarose gel electrophoresis (SDD-AGE) was performed according to a published protocol with a few modifications. Cells were lysed in NP40 lysis buffer (10 mM Tris-HCl, pH 7.5, 150 mM NaCl, 1 mM EDTA, pH 8.0, 1% NP40) containing fresh protease inhibitor cocktail on ice for 30 min and then frozen and thawed twice. The cells were then lysed by sonication. An equal amount of $1 \times$ sample loading buffer (0.5 × TBE, 10% glycerol, 2% SDS, 0.0025% bromophenol blue) was added to the lysate, and the mixture was loaded onto a vertical 1.5% agarose gel (containing 10% SDS). After electrophoresis in running buffer ($1 \times$ TBE and 0.1% SDS) for 35 min at a constant voltage of 100 V at 4 °C, the proteins were transferred to a nitrocellulose (NC) membrane for immunoblotting. For native page, the cell was resuspended by Lysis Buffer (50 mM Tris-HCL, pH 8.0, 150 mM NaCl, 1 mM PMSF, 1 mM EDTA, 1% NP40, 1 mM Na$_3$VO$_4$, 0.5% Sodium deoxycholate). It was vortexed for 15 s and then lysed on ice for about 60 min. Afterwards, it was freeze and thawed for thrice. The lysate was then centrifuged for 30 min at 15,000 rpm. Adding 5 μl 5 × Sample Loading Buffer (0.3 M Tris-HCL, pH 6.8, 50% glycerol, 0.0025% bromophenol blue) into the 20 μl lysate and the lysate was loaded onto a vertical polyacrylamide gel which was pre-electrophoresis for 30 min with a constant current of 30 mA. After electrophoresis in the running buffer (Tris base: 15.12 g, Glucine: 72.06 g, pH 8.47 for 500 mL 1 × Native Running Buffer) for 35 min with a constant current of 5 mA and 2 h with a constant current of 10 mA, the proteins were transferred to an PVDF membrane for immunoblotting.

**RNA isolation and real-time PCR**. Cells were lysed with TRIzol (TAKARA, Otsu, Japan), and RNA was isolated according to standard protocols. Total RNA was then used for reverse transcription according to the manufacturer's instructions (Fermentas, Vilnius, Lithuania). The mRNA was quantitated via quantitative PCR.

All real-time PCR values were normalized to *GAPDH* mRNA expression. The oligonucleotides used in the study are presented in Supplementary Table 1.

**In vitro kinase assays**. We incubated aliquots of the immunoprecipitants in 10 μL of kinase buffer comprising 10 mM HEPES (pH 7.4), 1 mM DTT, 5 mM $MgCl_2$, and 5 mCi [γ-$^{32}$P] ATP at 25 °C for 30 min. The samples were then resolved via SDS-PAGE, and phosphorylated proteins were visualized by autoradiography.

**Statistical analysis**. Data are expressed as the mean ± SEM. Statistical significance was evaluated using the unpaired two-tailed Student's *t*-test, and comparisons among more than two groups were performed using one-way or two-way ANOVA with Bonferroni post hoc or Tukey test post hoc analysis. Differences were considered significant at a *P*-value < 0.05.

**Reporting summary**. Further information on research design is available in the Nature Research Reporting Summary linked to this article.

## Data availability

The authors declare that all data supporting the findings of this investigation are available within the article, within its supplementary information, and from the corresponding authors upon reasonable request. The source data underlying Figs. 1a–e, 2a–d, g–i, 3a–e, h, 4a–c, e, f, 5a, b, d, e, 6a–e, 7a–g, and 8a–d, f–h, and Supplementary Figs. 1, 2, 3, and 4 are provided as a Source Data file.

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

## Acknowledgements

We thank Drs. Hongbing Shu, Min Wu, and Fenyong Liu for providing reagents, plasmids, viruses, and cell lines. This work was supported by grants from the National Natural Science

Foundation of China (81872271, 30971499, 81470375, and 81602450), the National Science and Technology Support Project (2014BAI02B00), the Trans-Century Training Program Foundation for the Talents by the State Education Commission (NCET-10-0655) and the Fundamental Research Funds for the Central Universities (204275771).

## Author contributions

S.-Z.L. and X.-D.Z. contributed to the experimental design; S.-Z.L., Q.-P.S., Y.S., H.-H.Z., Y.L., B.-X.J., T.-Z.L., C.L., and X.-C.H. performed the research; S.-Z.L., and X.-D.Z. analyzed the data; W.S. and B.Z. provided reagents and support for mice; S.-Z.L., H.-H. Z., R.-L.D., and X.-D.Z. obtained the funding; and S.-Z.L. and X.-D.Z. wrote the paper.

## Additional information

**Competing interests:** The authors declare no competing interests.

