## [Peer Review File · Nature Communications]

Reviewers' comments:

Reviewer #1 (Remarks to the Author):

A multilayered autoinhibitory mechanism that modulates MAVS activity in the absence of stimulation has recently been described, whereas mechanisms that regulate MAVS activity after stimulation still remain largely elusive. In order to identify kinases potentially regulating virus-induced MAVS signaling, the authors of this study screened approximately 100 different kinases in an IFN-beta reporter assay after Sendai virus (SeV) infection. In that assay nemo-like kinase (NLK) turned out to show an inhibitory effect. Therefore, in this study Li et al. investigated the mechanism of how NLK inhibited virus-induced MAVS activity. They found that NLK phosphorylates MAVS and thus inhibits its activity. Furthermore, they discovered that the MAVS peptide 471-480 (NAMDAP) is critically needed for the interaction between NLK and MAVS. Finally, in in vitro as well as mouse infection experiments NAMDAP treatment augmented antiviral immunity and enhanced protection.

In this study data are provided supporting the hypothesis that NLK diminishes MAVS activity following virus stimulation. However, what does this mean conceptually? NLK has been implemented in the regulation of several other biological processes before. Is NLK expression generally needed in order to avoid too high IFN-beta expression following virus infection, or would this mechanism be effective only under certain conditions and/or in certain cell subsets? An answer to this question is needed in order to understand the role of NLK in viral pathogenesis. Such questions could ideally be addressed in the conditional NLK mice that were generated in the context of this study.

The data presented in Figure 7e imply that the MAVS-derived peptide NAMDAP is needed for the interaction between NLK and MAVS. Furthermore, NAMDAP treatment of NLK expressing 293T cell showed enhanced IFN-beta expression following VSV infection. Is this effect mediated by NAMDAP binding to NLK that would inhibit the interaction between NLK and MAVS? In the current version of the manuscript the mechanism of how NAMDAP treatment enhances antiviral responses remains unclear. Unfortunately, the formulation in line 336 "we determined if NAMDAP could restore degraded MAVS" is misleading, or do the authors think of another mode of action of NAMDAP than inhibiting the interaction between NLK and MAVS?

Major points:

For none of the experiments presented in this study it is indicated how often the experiments were repeated. Therefore, the reproducibility of the effects cannot be estimated.

Important references are missing throughout the manuscript:

No reference is given for the Cre expressing mouse line *Lyz2-Cre* that was used to generate conditional *Nlk^{fl/fl}*-Cre mice.

No reference is given for the cell line HCT116/*NLK*^{-/-}.

No reference is given for the viruses used, neither for SeV nor for VSV or VSV-GFP.

In this study the VSV-GFP was used for in vitro as well as in vivo infection experiments. Previous reports revealed that GFP expressing VSV showed reduced pathogenesis when compared with the wild type virus. Thus, how is it possible that upon i.v. injection of 10E7 PFU of VSV-GFP a 100% lethal disease course is observed in wild type mice? Usually i.v. injection of 10E7 PFU of most VSV strains is tolerated by at least 70-90% of C57BL/6 mice. Details of the infection experiment have to be provided, otherwise it is difficult to understand the data.

Minor points:

The legend of Figure 2 is mislabeled. In that legend explanations are given for (f), whereas this should be (e).

In the FACS blots of Figure 4d and 7d labeling of the x-axis is missing.

Reviewer #2 (Remarks to the Author):

In this study, the authors report function for the kinase NLK in antiviral immunity. A mechanism is proposed whereby NLK promotes the degradation of the MAVS adaptor to prevent RLR signaling during viral infection. There are many concerns that I have, from a technical perspective, which are outlined below. These concerns limit enthusiasm for this study.

1. The fractionation experiments presented in Figure 1E are of poor quality and it is difficult to agree with the conclusion that NLK moves during infection. Much additional work would need to be done to validate this conclusion.

2. The plaque assay results presented in Figure 2F and 2G are rather modest, with only a few-fold change of virus replication. One would expect that a central regulator of antiviral signaling would result in a log-based phenotype in viral replication. It is therefore difficult to agree with the conclusion that NLK is a critical regulator of antiviral immunity.
3. My concerns in point#2 would be considered assuaged by the in vivo injections of VSV in Figure 2H and I, but these latter experiments also have problems. The authors show that viral plaque forming unit drops to zero within 48 hours of infection of both strains of mice. Despite this complete elimination of infection, the authors show that both strains of mice die 1-2 weeks later. Why are the mice dying if the virus has been eliminated?
4. The microscopy presented in Figure 5C is problematic, as MAVS-GFP is not a good reagent and does not signal properly. The authors are encouraged to perform similar studies using human cells where antibodies against the endogenous MAVS are suitable for microscopy.
5. The authors neglect to explore the localization of NLK to peroxisomes and mitochondria, both of which harbor signaling-competent MAVS. The authors are encouraged to study MAVS on both of these organelles for localization with NLK.
6. The interactions between MAVS and NLK increase upon infection. Based on knowledge of MAVS signaling, viral infection causes MAVS to oligomerize into a large prion-like signaling structure. The authors are encouraged to perform studies to determine if NLK is present in a complex with oligomeric MAVS or monomeric MAVS after viral infection.
7. The studies with the inhibitory peptide in Figure 7 are concerning. One reason for this concern is Figure 7E, where the authors show that, in cell culture, VSV reaches titers of $\sim 10^{12}$ pfu/ML. That is an unprecedented level of viral replication by several logs, and one that this referee is uncomfortable agreeing with.
8. Finally, the mechanism proposed would indicate that NLK deficient cells would have no defect in non-MAVS dependent activities. No such controls have been performed.

Point-by-point response to reviewers' comments

We would like to thank the reviewers for their comments and insightful suggestions, which have greatly helped us to revise the manuscript and improve our study. Based on the reviewers' comments, we have performed additional experiments and clarified certain statements/experimental procedures in the revised manuscript. Following is our point-by-point response to the reviewers' comments.

Reviewer #1

In this study data are provided supporting the hypothesis that NLK diminishes MAVS activity following virus stimulation. However, what does this mean conceptually? NLK has been implemented in the regulation of several other biological processes before. Is NLK expression generally needed in order to avoid too high IFN-beta expression following virus infection, or would this mechanism be effective only under certain conditions and/or in certain cell subsets? An answer to this question is needed in order to understand the role of NLK in viral pathogenesis. Such questions could ideally be addressed in the conditional NLK mice that were generated in the context of this study.

Response:

We thank the reviewer for his/her insightful comments. To address this question, we examined the role of NLK in antiviral signaling in different cell subsets. Firstly, we generated Lyz2-Cre $Nlk^{fl/fl}$ and Lyz2-Cre $Nlk^{+/+}$ BMDCs and infected these cells with SeV followed by qRT-PCR or ELISA analysis. The results showed that NLK was efficiently deleted in Lyz2-Cre $Nlk^{fl/fl}$ BMDCs and knockout of NLK potentiated SeV-induced expression of *Ifnb* and *Ccl5* (Fig 2b-c). Secondly, we isolated $Nlk^{fl/fl}$ and $Nlk^{+/+}$ MLFs, infected these cells with Ad-GFP or Ad-Cre for overnight and cultured these cells in normal medium for two days followed by SeV infection and qRT-PCR or ELISA analysis. The results showed that Ad-Cre infection led to ~50-60% reduction of NLK compared to Ad-GFP infection (Fig S1f). Such a reduction resulted in hyper-induction of *Ifnb* and *Ccl5* after SeV infection (Fig S1g-h). These data together suggest that NLK negatively regulates RNA virus-triggered signaling in multiple types of cells. Thus, it is likely that the expression of NLK is generally needed to avoid too high type I IFN expression during viral infection. On the other hand, such a negative regulation might be hijacked or used by viruses to inhibit antiviral responses and benefit for self-replication. We have included the information in the revised manuscript.

The data presented in Figure 7e imply that the MAVS-derived peptide NAMDAP is needed for the interaction between NLK and MAVS. Furthermore, NAMDAP treatment of NLK expressing 293T cell showed enhanced IFN-beta expression following VSV infection. Is this effect mediated by NAMDAP binding to NLK that would inhibit the interaction between NLK and MAVS? In the current version of the manuscript the mechanism of how NAMDAP treatment enhances antiviral responses remains unclear. Unfortunately, the formulation in line 336 “we determined if NAMDAP could restore degraded MAVS” is misleading, or do the authors think of another mode of action of NAMDAP than inhibiting the interaction between NLK and MAVS?

Response:

The question raised by the reviewer is quite interesting and important for us to understand the mechanism in depth. We have examined the effect of NAMDAP on NLK-MAVS association in the presence or absence of SeV infection. The results showed that SeV-induced association of NLK-MAVS was substantially inhibited by the peptide NAMDAP (Fig 8b). In addition, SeV-induced degradation of MAVS was inhibited by NAMDAP treatment (Fig S4c), indicating that NAMDAP blocks the interaction of NLK-MAVS and rescued MAVS from degradation after viral infection.

Thank the reviewer for pointing the mistake (original line 336) for us, we have changed the statement into “Thus, we also determined if NAMDAP protect MAVS from degradation induced by interaction between NLK and MAVS”. (page 17).

Major points:

For none of the experiments presented in this study it is indicated how often the experiments were repeated. Therefore, the reproducibility of the effects cannot be estimated.

Response:

We are very confident about all the results. Most of the experiments were performed at least 3 times. Especially, for some critical key experiments, we asked different people to perform and reach 5-6 times. Actually, we found this molecular mechanism starting from 2012, and we spend long time to get the conditional KO mice. we can guarantee all the important experiments can be easily repeated. And we described the repeated time in the figure legends.

Important references are missing throughout the manuscript:

No reference is given for the Cre expressing mouse line Lyz2-Cre that was used to generate conditional Nlkl/flLyz2-Cre mice.

Response:

The Lyz2-Cre mice were from Jackson Lab (B6.129P2-Lyz2^{tm1(cre)Ifo}/J, Stock No: 004781) and used as previously described (Liuyu T et al., Cell Res, 2019, 29(1): 67-79). The related information has been included in the Methods. We have already added the reference into the section of Cell Lines, Retroviral Gene Transfer and Mice. Please check it in this section.

No reference is given for the cell line HCT116/NLK-/-.

Response:

NLK^{-/-} HCT116 cells were generated by a two-step rAAV-mediated gene targeting method (Li SZ et al., Biochim Biophys Acta., 2014, 1843(7):1365-72.). We have already added citations in the manuscript (page 9).

No reference is given for the viruses used, neither for SeV nor for VSV or VSV-GFP.

Response:

These reagents were previously described (Lin D et al., PNAS, 2015, 112(36): 11324-9; Liuyu T et al., Cell Res, 2019, 29(1): 67-79). The related references have been cited in the section of Reagents and Constructs (page 22).

In this study the VSV-GFP was used for in vitro as well as in vivo infection experiments. Previous reports revealed that GFP expressing VSV showed reduced pathogenesis when compared with the wild type virus. Thus, how is it possible that upon i.v. injection of 10E7 PFU of VSV-GFP a 100% lethal disease course is observed in wild type mice? Usually i.v. injection of 10E7 PFU of most VSV strains is tolerated by at least 70-90% of C57BL/6 mice. Details of the infection experiment have to be provided, otherwise it is difficult to understand the data.

Response:

We do agree with the reviewer that GFP insertion into the genome of VSV attenuates the pathogenesis. We observed that i.v. injection of 10^7 PFU VSV-GFP could not lead to death in mice and the VSV-GFP was cleared within 48 hours after injection (original Fig 2h). We thus used wild-type VSV (no GFP insertion) to examine the in vivo role of NLK in antiviral responses (original Fig 2i). However, injection of 10^8 PFU VSV resulted in quick death of 8 week-old Lyz2-Cre $Nlk^{fl/fl}$ and $Nlk^{fl/fl}$ mice. We titrated viral titers and mice ages and found that i.v. injection of 2×10^7 PFU VSV led to 60-80% death for 8-10 week-old mice (Liuyu T et al., Cell Res, 2019, 29(1): 67-79; Lu B et al., Cell Host Microbe, 2017, 22(1):86-98), and 100% death for 6-7 week-old mice (this study). As shown in Fig 2, the $Nlk^{fl/fl}$ mice started to die at day 3 after VSV infection and all of the mice died at day 8 after infection. In contrast, the Lyz2-Cre $Nlk^{fl/fl}$ mice started to die at day 4 after VSV infection and one out of six mice survived at the end point of the experiments, indicating that NLK negatively regulates antiviral responses in vivo. The related information has been included in the figure legends or methods.

Minor points:

The legend of Figure 2 is mislabeled. In that legend explanations are given for (f), whereas this should be (e).

Response:

Thank the reviewer for point this out for us. We have corrected the mistake.

In the FACS blots of Figure 4d and 7d labeling of the x-axis is missing.

Response:

The x-axis in original Fig 4d and 7d should be GFP, which has been included.

Reviewer #2

1.The fractionation experiments presented in Figure 1E are of poor quality and it is

difficult to agree with the conclusion that NLK moves during infection. Much additional work would need to be done to validate this conclusion.

Response:

We agree with the reviewer that results from the fractionation experiments did not support the conclusion that NLK moves from the membrane fraction to mitochondria (original Fig 1e). To further examine the movement of NLK after viral infection, we performed immunofluorescence and confocal microscopy analysis. The results showed that NLK was constitutively colocalized with mitotracker in the absence of viral infection, and SeV infection slightly increase the colocalization of NLK-mitotracker (Fig 1f), indicating that NLK did accumulate on mitochondria after viral infection.

It has been reported that both peroxisomes and mitochondria could as the antiviral platform (Dixit E et al., Cell, 2010, 141(4): 668-81). Interestingly, we found that a small fraction of NLK was colocalized with the peroxisome marker Catalase. SeV infection also substantially increased the colocalization of NLK-Catalase at 6 hours and at 9 hours, respectively (Fig 1g). Consistent with these observations, results from differential centrifugation and gradient centrifugation assays suggested that the amount of NLK in the peroxisome and mitochondria fraction was substantially increased (Fig 1e).

In our study, we found that SeV-induced expression of immediate early antiviral genes and type I IFNs was potentiated in NLK knockout cells (Fig 2), indicating that NLK might regulate MAVS at both peroxisomes and mitochondria. Sure, further investigations are required to address how NLK is accumulated on mitochondria and peroxisomes in response to viral infection.

2. The plaque assay results presented in Figure 2F and 2G are rather modest, with only a few-fold change of virus replication. One would expect that a central regulator of antiviral signaling would result in a log-based phenotype in viral replication. It is therefore difficult to agree with the conclusion that NLK is a critical regulator of antiviral immunity.

Response:

We agree with the reviewer that knockout of NLK in human cell lines resulted in log-scale difference (Fig 3, 10^8 v.s. $b \times 10^7$ PFU/ml for NLK^{+/+} and NLK^{-/-} HCT116 cells, respectively). According to our experience, VSV infection (MOI=1) in HEK293 or HeLa cells for 24 hours would result in 10^6 - 10^8 PFU/ml. Therefore, the differences

of VSV plaques in control and knockdown or knockout cell lines usually reach log scales (Zhong B et al., *Immunity*, 2008, 29(4): 538-50; Zhong B et al., *Immunity*, 30(3):397-407; Huang J et al., *EMBO J*, 2005, 24(23):4018-28).

In primary mouse cells (BMDMs or BMDCs), however, a MOI of 4 or higher for 12-24 hours results in severe cell death (the VSV in the supernatant is usually at 10^4 PFU/ml scale) and a MOI of 0.5 or lower for 12-24 hours yields undetectable viral particles ($<10^2$ PFU/ml) (Liuyu T et al., *Cell Res*, 2019, 29(1): 67-79; Lu B et al., *Cell Host Microbe*, 2017, 22(1):86-98). In this study, we used a MOI of 4 of VSV to infect Lyz2-Cre $Nik^{fl/fl}$ and $Nik^{fl/fl}$ BMDMs. One hour later, the virus was removed and the cells were washed twice with prewarmed PBS and cultured with 250 μ l fresh full medium for 18 hours. The supernatants were harvested and serially diluted for standard plaque assays. The results showed that there are 2.2×10^4 PFU/ml and 5.0×10^3 PFU/ml in the cultured supernatants of $Nik^{fl/fl}$ and Lyz2-Cre $Nik^{fl/fl}$ BMDMs (Fig 2f-g). Although the difference of VSV particles in the supernatants of Lyz2-Cre $Nik^{fl/fl}$ and $Nik^{fl/fl}$ BMDMs is sub-log scale, this difference is reproducible and statistically significant.

3. My concerns in point#2 would be considered assuaged by the in vivo injections of VSV in Figure 2H and I, but these latter experiments also have problems. The authors show that viral plaque forming unit drops to zero within 48 hours of infection of both strains of mice. Despite this complete elimination of infection, the authors show that both strains of mice die 1-2 weeks later. Why are the mice dying if the virus has been eliminated?

Response:

We are sorry for not describing the experiments clearly. Firstly, for our experiment, we injected VSV-GFP virus at 10^7 in fig 2h to collect the serum and wild-type VSV (2×10^7 PFU) and younger mice (6-7 weeks) in Fig 2i for survival assay. You could find the description in the figure legends 2h and 2i. These are two separate experiments.

Secondly, VSV-GFP are cleared within 48-72 hours after i.v. injection (even as high as 10^8 PFU VSV-GFP viruses were used) is consistent with Sun's paper (Sun Q et al., *Immunity*, 2006, 24(5):633-42). Actually, we experience lots of times low titer virus in the serum and always cleared within 48-72 hours after i.v. injection. Thus, we are also try to explain it due to curiosity one year ago. One of the potential hypotheses is that the virus still stay in brain and the other organs. We take the mice brain after virus infection for 4 days. The homogenized brain was serially diluted and subject to standard plaque assays. The results showed that the brain still have 3×10^3 PFU/g virus. Thus, we believe that virus in brain give rise to the mice death.

4. The microscopy presented in Figure 5C is problematic, as MAVS-GFP is not a good reagent and does not signal properly. The authors are encouraged to performed similar studies using human cells where antibodies against the endogenous MAVS are suitable for microscopy.

Response:

Following the reviewer's suggestion, we performed immunofluorescence and confocal microscopy analysis. The results showed that few NLK was colocalized with MAVS without infection, and SeV infection obviously induced colocalization of NLK and MAVS (Fig 5c). The related information has been included in the text.

5. The authors neglect to explore the localization of NLK to peroxisomes and mitochondria, both of which harbor signaling-competent MAVS. The authors are encouraged to study MAVS on both of these organelles for localization with NLK.

Response:

Study from Dr. Jonathan Kagan's group has elegantly demonstrated that MAVS is localized on both mitochondria and peroxisomes. The NLK-MAVS localization was proved and which substantially increased by SeV infection (Fig 5c), indicated that NLK may have both mitochondria and peroxisomes localization. We further constructed mitochondria localized MAVS (mito), peroxisome localized MAVS (Per) and cytosol localized MAVS (Cyto). NLK could interact with both peroxisomes or mitochondria MAVS but not with cytosol localized MAVS (revised fig. 5d). It is highly possible that both mitochondrial and peroxisomal MAVS are associated with and regulated by NLK.

6. *The interactions between MAVS and NLK increase upon infection. Based on knowledge of MAVS signaling, viral infection causes MAVS to oligomerize into a large prion-like signaling structure. The authors are encouraged to perform studies to determine if NLK is present in a complex with oligomeric MAVS or monomeric MAVS after viral infection.*

Response:

We thank the reviewer for his/her insightful comments. To address this question, we harvested cells in the presence or absence of SeV infection and performed SDD-AGE and immunoblot assays as previously described. Consistent with previous studies, SeV infection results in aggregation of MAVS. Interestingly, NLK was also detected as aggregates (Fig5e). In our immunofluorescence and confocal microscopy assays, we found that NLK was colocalized with MAVS after viral infection (Fig5c). Therefore, it is likely that NLK is present in a complex with oligomeric MAVS. The related information has been included in the text.

7. *The studies with the inhibitory peptide in Figure 7 are concerning. One reason for this concern is Figure 7E, where the authors show that, in cell culture, VSV reaches titers of $\sim 10E12$ pfu/ML. That is an unprecedented level of viral replication by several logs, and one that this referee is uncomfortable agreeing with.*

Response:

We are sorry for mislabeling the figure that confused the reviewer. The numbers of PFU/ml were 7.5×10^8 and 2.25×10^8 for control and NAMDAP groups, respectively. The correct log numbers should be $\log(7.5 \times 10^8) = 8.875$ and $\log(2.25 \times 10^8) = 8.339$, respectively. We have mistakenly calculated as $\log(75) \times 7 = 13.1$ and $\log(22.5) \times 7 = 9.5$. These mistakes have been corrected and the old Fig 7e was replaced by a new one.

8. *Finally, the mechanism proposed would indicate that NLK deficient cells would have no defect in non-MAVS dependent activities. No such controls have been performed.*

Response:

Following the reviewer's suggestion, we have examined the role of NLK in cytoplasmic dsDNA-induced expression of downstream genes. As shown in Fig s3c-d, knockout of NLK in BMDMs did not affect transfected DNA ligands (such as ISD45,

HSV60, DNA90 and HSV120)-induced expression of *Ifnb* and *Ccl5*, indicating that NLK is not involved in cytoplasmic DNA-triggered induction of downstream genes. The related information has been included in the text.

c

d

REVIEWERS' COMMENTS:

Reviewer #1 (Remarks to the Author):

The authors undertook significant effort in order to address the reviewers' concerns. In my view the manuscript significantly improved.

Reviewer #2 (Remarks to the Author):

In this revised study, the authors have done a good job addressing my prior concerns. A few mistakes were corrected and new experiments verified and expanded the scope of the conclusions. I have no additional experimental suggestions. However, the authors are encouraged to make several changes to the text, for clarity and accuracy.

1. The reviews cited are extremely outdated. References 1-3 range from 9-13 years old and these reviews are presented by the authors are representative of our current knowledge. Please replace these citations with more current literature.

2. Same criticism for references 4-6, all of which are reviews.

3. Reference 11 was published contemporaneously with the following reference: Nat Immunol. 2009 Oct;10(10):1065-72. Both references should be cited.

4. Reference 22 is cited in the introduction incorrectly. Please adjust.

5. References to the new data presented on peroxisomes are not present anywhere in the text. Please adjust.

REVIEWERS' COMMENTS:

Reviewer #1 (Remarks to the Author):

The authors undertook significant effort in order to address the reviewers' concerns. In my view the manuscript significantly improved.

Reviewer #2 (Remarks to the Author):

In this revised study, the authors have done a good job addressing my prior concerns. A few mistakes were corrected and new experiments verified and expanded the scope of the conclusions. I have no additional experimental suggestions. However, the authors are encouraged to make several changes to the text, for clarity and accuracy.

1. The reviews cited are extremely outdated. References 1-3 range from 9-13 years old and these reviews are presented by the authors are representative of our current knowledge. Please replace these citations with more current literature.

Response:

We have already replaced these citations with more current literature.

2. Same criticism for references 4-6, all of which are reviews.

Response:

We have already replaced these citations with more current literature.

3. Reference 11 was published contemporaneously with the following reference: Nat Immunol. 2009 Oct;10(10):1065-72. Both references should be cited.

Response:

We have already added this reference together with the original reference 11.

4. Reference 22 is cited in the introduction incorrectly. Please adjust.

Response:

We replaced the original reference 22 with the revised reference 21.

5. References to the new data presented on peroxisomes are not present anywhere in the text. Please adjust.

Response:

We added the revised reference 37 to demonstrate the crucial role of peroxisomes in innate immunity.

Finally, we thank the reviewers for their suggestions and positive comments, which helped to make our story more complete.